# VIPaint: Image Inpainting with Pre-Trained Diffusion Models via Variational Inference

## Abstract

Diffusion probabilistic models learn to remove noise that is artificially added to the data during training. Novel data, like images, may then be generated from Gaussian noise through a sequence of denoising operations. While this Markov process implicitly defines a joint distribution over noise-free data, it is not simple to condition the generative process on masked or partial images. A number of heuristic sampling procedures have been proposed for solving inverse problems with diffusion priors, but these approaches do not directly approximate the true conditional distribution imposed by inference queries, and are often ineffective for large masked regions. Moreover, many of these baselines cannot be applied to latent diffusion models which use image encodings for efficiency. We instead develop a hierarchical variational inference algorithm that analytically marginalizes missing features, and uses a rigorous variational bound to optimize a non-Gaussian Markov approximation of the true diffusion posterior. Through extensive experiments with both pixel-based and latent diffusion models of images, we show that our VIPaint method significantly outperforms previous approaches in both the plausibility and diversity of imputations, and is easily generalized to other inverse problems like deblurring and superresolution.

## 1 Introduction

Diffusion models (Ho et al., 2020b; Song et al., 2021b; Nichol & Dhariwal, 2021; Song & Ermon, 2019) and hierarchical *variational autoencoders* (VAEs) (Child, 2021; Vahdat & Kautz, 2020; Sønderby et al., 2016) are generative models in which a sequence of latent variables encode a rich data representation. For diffusion models, this latent structure is defined by a diffusion process that corrupts data over "time" via additive Gaussian noise. While each step of hierarchical VAE training requires end-to-end inference of all latent variables, diffusion models estimate stochastic gradients by sampling a few timesteps, and learning to incrementally denoise corrupted data. Given a learned denoising network, synthetic data is generated by sequentially refining Gaussian noise for hundreds or thousands of time steps, producing deep generative models that have advanced the state-of-the-art in natural image generation (Dhariwal & Nichol, 2021; Kingma et al., 2021a; Karras et al., 2022).

Diffusion models for high-dimensional data like images are computationally intensive. Efficiency may be improved by leveraging an autoencoder (Kingma & Welling, 2019; Rombach et al., 2022b; Vahdat et al., 2021) to map data to a lower-dimensional encoding, and then training a diffusion model for the lower-dimensional codes. This dimensionality reduction enables tractable but expressive models for images with millions of pixels. The effectiveness of *latent diffusion models* (LDMs) has made them a new standard for natural image generation, and they are thus our focus here.

Motivated by the foundational information captured by diffusion models of images, numerous algorithms have incorporated a pre-trained diffusion model as a prior for image editing (Meng et al., 2021), inpainting (Song et al., 2021b; Wang et al., 2023b; Kawar et al., 2022; Chung et al., 2022b; Lugmayr et al., 2022; Cardoso et al., 2024; Feng et al., 2023; Trippe et al., 2023; Dou & Song, 2024), or other inverse problems (Kadkhodaie & Simoncelli, 2021; Song et al., 2023; Graikos et al., 2022; Mardani et al., 2023; Chung et al., 2023). Many of these prior methods are specialized to inpainting with pixel-based diffusion models, where every data dimension is either perfectly observed or completely missing, and are not easily adapted to state-of-the-art LDMs.

Figure 1: VIPaint inpainting with a pretrained, unconditional LDM (Rombach et al., 2022b) of LSUN churches. For two image-mask pairs (left columns), we show the expected reconstruction for a sample taken from our hierarchical VIPaint posterior at times $\{T_e = 550, 500, 450, T_s = 400\}$. We skip intermediate noise levels between these critical times during variational optimization, and add fine-grained details to our final inpaintings (three right columns, Inpainting 1 corresponds to $t = 200$ samples via 400 sequential denoising steps at times $0 \leq t < T_s$.

Most algorithms for inpainting with diffusion models employ an iterative refinement procedure, like that used to generate unconditional samples, and then guide their predictions towards the partially observed image via various approximations and heuristics. But by sequentially annealing from independent Gaussian noise to noise-free images, these approaches produce myopic samples that do not adequately incorporate information from observed pixels, and fail to correct errors introduced in earlier stages of the "reverse-time" diffusion. More recent work has extended these approaches to enable image editing (Avrahami et al., 2022) or inpainting (Rout et al., 2023; Corneanu et al., 2024; Chung et al., 2023; Song et al., 2024) with LDMs, but continues to suffer similar inaccuracies.

In this work, we present *VIPaint*, a novel application of *variational inference* (VI) (Wainwright & Jordan, 2008; Blei et al., 2017) that efficiently optimizes a hierarchical, Markovian, non-Gaussian approximation to the true LDM posterior. VI has achieved excellent image restoration results with a wide range of priors, including mixtures (Fergus et al., 2006; Ji et al., 2017) and hierarchical VAEs (Agarwal et al., 2023), but there is little work exploring its integration with state-of-the-art LDMs. While *Red-Diff* (Mardani et al., 2023) applies VI to approximate the posterior of pixel-based DMs, its local approximation of the noise-free image posterior is difficult to optimize, requiring annealing heuristics that we demonstrate are sensitive to local optima. Our VIPaint method instead defines a hierarchical posterior that strategically accounts for a subset of noise levels, enabling the inference of both high-level semantics and low-level details from observed pixels *simultaneously* (see Fig. 1). We efficiently infer variational parameters via non-amortized optimization for each inpainting query, avoiding the need to collect a training set of corrupted images (Liu et al., 2024; Corneanu et al., 2024), expensively fine-tune generative models (Avrahami et al., 2022) or variational posteriors (Feng et al., 2023) for each query, or retrain large-scale conditional diffusion models (Rombach et al., 2022b; Saharia et al., 2022; Nichol et al., 2022; Chung et al., 2022a).

We begin by reviewing properties of (latent) diffusion models in Sec. 2, and prior work on inferring images via pre-trained diffusion models in Sec. 3. Sec. 4 then develops the VIPaint algorithm, which first fits a hierarchical posterior that best aligns with the observations, and then samples from this approximate posterior to produce diverse reconstruction hypotheses. Results in Sec. 5 on inpainting, and the Appendix on other inverse problems, then show substantial qualitative and quantitative improvements in capturing multimodal uncertainty for both pixel-based and latent DMs.

## 2 BACKGROUND: DIFFUSION MODELS

The diffusion process begins with clean data $x$, and defines a sequence of increasingly noisy versions of $x$, which we call the *latent variables* $z_t$, where $t$ runs from $t = 0$ (low noise) to $t = T$ (substantial noise). The distribution of latent variable $z_t$ conditioned on $x$, for any integer time $t \in [0, T]$, is

$$q(z_t \mid x) = \mathcal{N}(z_t \mid \alpha_t x, \sigma_t^2 I), \tag{1}$$

where $\alpha_t$ and $\sigma_t$ are strictly positive scalar functions of $t$. This noise implicitly defines a Markov chain for which the conditional $q(z_t \mid z_{t-1})$ is also Gaussian. Also, $q(z_{t-1} \mid z_t, x)$ is Gaussian (see Appendix B.1) with mean equal to a linear fuction of the input data $x$ and the latent sample $z_t$.

The signal-to-noise ratio (Kingma et al., 2021b) induced by this diffusion process at time $t$ equals $SNR(t) = \alpha_t^2/\sigma_t^2$. The SNR monotonically decrease with time, so that $SNR(t) < SNR(s)$ for $t > s$. Diffusion model performance is very sensitive to the rate at which SNR decays with time,

or equivalently the distribution with which times are sampled during training (Nichol & Dhariwal, 2021; Karras et al., 2022). This DM specification includes variance-preserving diffusions (Ho et al., 2020a; Sohl-Dickstein et al., 2015) as a special case, where $\alpha_t = \sqrt{1 - \sigma_t^2}$. Another special case, variance-exploding diffusions (Song & Ermon, 2019; Song et al., 2021b), takes $\alpha_t = 1$.

**Image Generation.** The generative model reverses the diffusion process outlined in Eq. (1), resulting in a hierarchical generative model that samples a sequence of latent variables $z_t$ before sampling $x$. Generation progresses backward in time from $t = T$ to $t = 0$ via a finite temporal discretization into $T \approx 1000$ steps, either uniformly spaced as in discrete diffusion models (Ho et al., 2020a), or via a possibly non-uniform discretization (Karras et al., 2022) of an underlying continuous-time stochastic differential equation (Song et al., 2021b). Denoting $t - 1$ as the timestep preceding $t$, for $0 < t < T$, the hierarchical generative model for data $x$ is expressed as follows:

$$p(x) = \int_z p(z_T)p(x \mid z_0) \prod_{t=1}^{T} p(z_{t-1} \mid z_t) \, dz. \tag{2}$$

The marginal distribution of $z_T$ is typically a spherical Gaussian $p(z_T) = \mathcal{N}(z_T \mid 0, \sigma_T^2 I)$. Pixel-based diffusion models take $p(x \mid z_0)$ to be a simple factorized likelihood for each pixel in $x$, while LDMs define $p(x \mid z_0)$ using a decoder neural network. The conditional latent distribution maintains the same form as the forward conditional distributions $q(z_{t-1} \mid z_t, x)$, but with the original data $x$ substituted by the output of a parameterized denoising model $z_0$ as

$$p_\theta(z_{t-1} \mid z_t) = q(z_{t-1} \mid z_t, z_0 = \hat{z}_\theta(z_t, t)), \quad \text{where} \quad \hat{z}_\theta(z_t, t) = \frac{z_t - \sigma_t \hat{\epsilon}_\theta(z_t, t)}{\alpha_t}. \tag{3}$$

This denoising model $\hat{\epsilon}_\theta(z_t, t)$ typically uses variants of the UNet architecture (Ronneberger et al., 2015) to predict the noise-free latent $z_0$ from its noisy counterpart $z_t$.

The Gaussian diffusion implies that $p_\theta(z_{t-1} \mid z_t) = \mathcal{N}(z_{t-1} \mid c_1(t)z_t + c_2(t)\hat{z}_\theta(z_t, t), \tilde{\sigma}_{t-1}^2 I)$, so the mean is a linear combination of the latent $z_t$ and the prediction $\hat{z}_\theta$, with constants determined from the diffusion hyperparameters as detailed in Appendix B.1. Our VIPaint approach flexibly accommodates multiple parameterizations of the denoising model, including the EDM model's direct prediction of $z_0$ for higher noise levels (Karras et al., 2022).

**Training Objective.** The variational lower bound (VLB) of the marginal likelihood is given by

$$- \log p(x) \leq \underbrace{-\mathbb{E}_{q(z_0|x)}[\log p_\theta(x|z_0)]}_{\text{reconstruction loss}} + \underbrace{D\left[q(z_T|z_0)||p(z_T)\right]}_{\text{prior loss}} + \underbrace{\mathcal{L}_{(0,T)}(z_0)}_{\text{diffusion loss}}, \tag{4}$$

where $D$ is the Kullback-Leibler (KL) divergence. The reconstruction loss, usually L1, can be estimated stochastically and differentiably using standard reparametrization techniques (Kingma & Welling, 2019). The prior loss is a constant because $p(z_T)$ is a Gaussian with fixed parameters. Ho et al. (2020b) express the diffusion loss for finite time $T$ as follows:

$$\mathcal{L}_{(0,T)}(z_0) = \sum_{t=1}^{T} \mathbb{E}_{q(z_t|z_0)} D\left[q(z_{t-1}|z_t, z_0)||p_\theta(z_{t-1}|z_t)\right]. \tag{5}$$

To boost training efficiency, instead of summing the loss over all $T$ times, timesteps are sampled from a uniform distribution $t \sim \mathcal{U}\{1, T\}$ to yield an *unbiased* approximation. Most prior work (Ho et al., 2020b; Song et al., 2021b) also chooses to optimize a re-weighted KL divergence that reduces sensitivity towards losses at very-low noise levels, so the final loss $\mathcal{L}_{(0,T)}(z_0)$ becomes

$$\mathcal{L}_{(0,T)}(z_0) = \frac{T}{2}\mathbb{E}_{\epsilon \sim \mathcal{N}(0,1), t \sim \mathcal{U}(1,T)}\left[||\epsilon - \hat{\epsilon}_\theta(z_t, t)||_2^2\right]. \tag{6}$$

**Latent Diffusion Models.** To encourage resource-efficient diffusion models, Rombach et al. (2022b); Vahdat et al. (2021) utilize an encoder $q_\phi(z_0|x)$ to map high-dimensional data $\mathbb{R}^D$ into a lower-dimension space $\mathbb{R}^d$ ($d < D$), and a decoder $p_\psi(x|z_0)$ to (approximately) invert this mapping. Together with an L1 reconstruction loss, the training loss for the autoencoder employs a combination of the perceptual loss (Zhang et al., 2018) and a patch-based adversarial objective (Rombach et al., 2022a) to encourage realism and reduce blurriness. Given this autoencoder, one can train a diffusion model in the space of low-dimensional encodings. The diffusion process is the same as defined in Eq. (1), but now corrupts $z_0 \sim q_\phi(z_0 \mid x)$ samples in the lower-dimensional space. Generation uses the reverse diffusion process to sample from $p_\theta(z_0)$ via the time-dependent noise prediction function $\hat{\epsilon}_\theta(z_t, t)$, and the decoder $p_\psi(x \mid z_0)$ to map the synthesized encodings $z_0$ to data space.

## 3 BACKGROUND: INFERENCE USING DIFFUSION MODELS

### 3.1 GENERAL INVERSE PROBLEMS

In many real-life scenarios, we encounter partial observations $y$ derived from an underlying $x$. Typically, these observations are modeled as $y = f(x) + v$, where $f$ represents a known linear degradation model and $v$ is Gaussian noise with $v \sim \mathcal{N}(0, \sigma_v^2)$. For instance, in an image inpainting task, $y$ might represent a *masked* imaged $y = x \odot m$, where $m$ is a binary mask indicating missing pixels.

In cases where the degradation of $x$ is significant, exactly recovering $x$ from $y$ is challenging, because many $x$ could produce the same observation $y$. To express the resulting posterior $p(x \mid y)$ given a DM prior, we can adapt the Markov generative process in Eq. (2) as follows:

$$p_\theta(x \mid y) = \int_z p_\theta(z_T \mid y) p_\theta(x \mid z_0, y) \prod_{t=1}^{T} p_\theta(z_{t-1} \mid z_t, y) \, dz. \tag{7}$$

Exactly evaluating this predictive distribution is infeasible due to the non-linear noise prediction (and decoder) network, and the intractable posteriors of latent codes $p(z_{t-1} \mid z_t, y)$ for all $t$.

**Blended** methods like Song et al. (2022); Wang et al. (2023a) define a procedural, heuristic approximation to the posterior and is tailored for image inpainting. They first generate unconditional samples $z_{t-1}$ from the prior using the learned noise prediction network, and then incorporate $y$ by replacing the corresponding dimensions with the observed measurements. RePaint Lugmayr et al. (2022) attempts to reduce visual inconsistencies caused by blending via a resampling strategy. A "time travel" operation is introduced, where images from the current time step $z_{t-1}$ are first blended with the noisy version of the observed image $y_{t-1}$, and then used to generate images in the $(t-1) + r, (r \geq 1)$ time step by applying a one-step forward process and following the Blended denoising process.

**Sampling Methods.** Motivated by the goal of addressing more general inverse problems, Diffusion Posterior Sampling (*DPS*) (Chung et al., 2023) uses Bayes' Rule to sample from $p_\theta(z_{t-1}|z_t, y) \propto p_\theta(z_{t-1}|z_t) p_\theta(y|z_{t-1})$. Instead of directly blending or replacing images with noisy versions of the observation, DPS uses the gradient of the likelihood $\log p_\theta(y|z_t)$ to guide the generative process at every denoising step $t$. Since computing $\nabla_{z_t} \log p(y|z_{t-1})$ is intractable due to the integral over all possible configurations of $z_{t'}$ for $t' < t - 1$, DPS approximates $p(y|z_{t-1})$ using a one-step denoised prediction $\hat{x}$ using Eq. (3). The likelihood $p(y|x) = \mathcal{N}(f(x), \sigma_v^2)$ can then be evaluated using these approximate predictions. To obtain the gradient of the likelihood term, DPS require backpropagating gradients through the denoising network used to predict $\hat{x}$.

Specializing to image inpainting, *CoPaint* (Zhang et al., 2023) augments the likelihood with another regularization term to generate samples $z_{t-1}$ that prevent taking large update steps away from the previous sample $z_t$, in an attempt to produce more coherent images. Further, it proposes CoPaint-TT, which additionally uses the time-travel trick to reduce discontinuities in sampled images.

Originally designed for pixel-space diffusion models, it is difficult to adopt these works directly to latent diffusion models. Posterior Sampling with Latent Diffusion (*PSLD*) (Rout et al., 2023) first showed that employing *DPS* directly on latent space diffusion models produces blurry images. It proposes to add another "gluing" term to the measurement likelihood which penalizes samples $z_t$ that do not lie in the encoder-decoder shared embedding space. However, this may produce artifacts in the presence of measurement noise (see Song et al. (2024)). To address this issue, recent concurrent work on the *ReSample* (Song et al., 2024) method divides the timesteps in the latent space into 3 subspaces, and optimizes samples $z_t$ in the mid-subspace to encourage samples that are more consistent with observations. Other work (Yu et al., 2023) highlights a 3-stage approach where data consistency can be enforced in the latter 2 stages which are closer to $t = 0$.

### 3.2 RED-DIFF: VARIATIONAL INFERENCE VIA FEATURE POSTERIORS

*RedDiff* (Mardani et al., 2023) approximates the true complex posterior $p(x \mid y)$ (Eq. 7) by a simple Gaussian distribution $q_\lambda(x) = \mathcal{N}(\mu, \sigma^2)$, where $\lambda = \{\mu, \sigma\}$ represents the variational parameters. Minimizing the KL divergence $D(q_\lambda(x)||p(x|y))$ guides the distribution $q$ to seek the *mode* in the

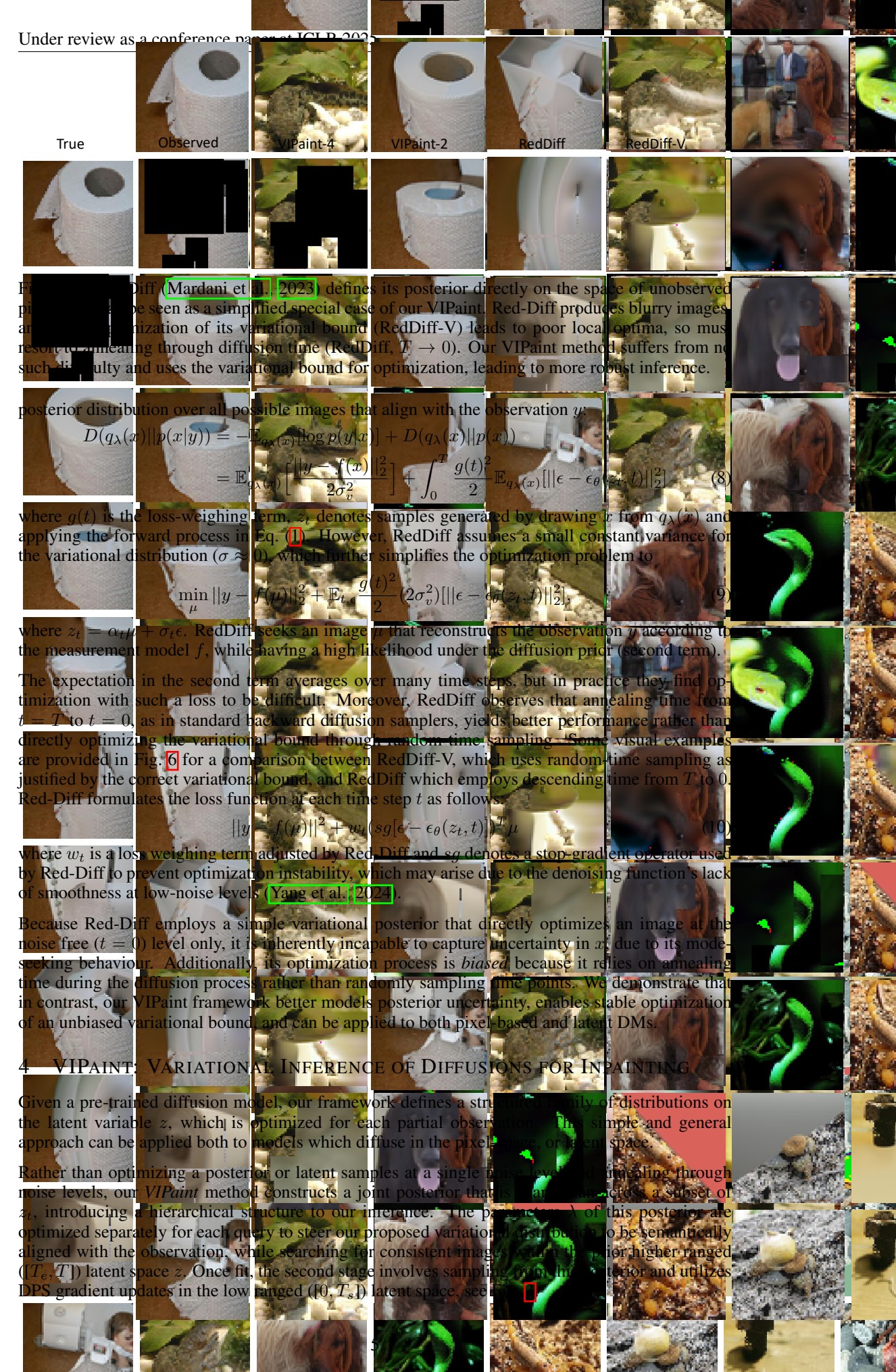

True     Observed     VIPaint-4     VIPaint-2     RedDiff     RedDiff-V

Fi... ...Diff (Mardani et al., 2023) defines its posterior directly on the space of unobserved pi... ...be seen as a simplified special case of our VIPaint. Red-Diff produces blurry images an... ...mization of its variational bound (RedDiff-V) leads to poor local optima, so mus... resort t... annealing through diffusion time (RedDiff, $T \to 0$). Our VIPaint method suffers from no such di... ...ulty and uses the variational bound for optimization, leading to more robust inference.

posterior distribution over all possible images that align with the observation $y$:

$$D(q_\lambda(x)||p(x|y)) = -\mathbb{E}_{q_\lambda(x)}[\log p(y|x)] + D(q_\lambda(x)||p(x))$$

$$= \mathbb{E}_{q_\lambda(x)}\left[\frac{||y - f(x)||_2^2}{2\sigma_v^2}\right] + \int_0^T \frac{g(t)^2}{2}\mathbb{E}_{q_\lambda(x)}[||\epsilon - \epsilon_\theta(z_t, t)||_2^2] \quad (8)$$

where $g(t)$ is the loss-weighing term, $z_t$ denotes samples generated by drawing $x$ from $q_\lambda(x)$ and applying the forward process in Eq. (1). However, RedDiff assumes a small constant variance for the variational distribution ($\sigma \approx 0$), which further simplifies the optimization problem to

$$\min_\mu ||y - f(\mu)||_2^2 + \mathbb{E}_t \frac{g(t)^2}{2}(2\sigma_v^2)[||\epsilon - \epsilon_\theta(z_t, t)||_2^2], \quad (9)$$

where $z_t = \alpha_t\mu + \sigma_t\epsilon$. RedDiff seeks an image $\mu$ that reconstructs the observation $y$ according to the measurement model $f$, while having a high likelihood under the diffusion prior (second term).

The expectation in the second term averages over many time-steps, but in practice they find optimization with such a loss to be difficult. Moreover, RedDiff observes that annealing time from $t = T$ to $t = 0$, as in standard backward diffusion samplers, yields better performance rather than directly optimizing the variational bound through random-time sampling. Some visual examples are provided in Fig. 6 for a comparison between RedDiff-V, which uses random-time sampling as justified by the correct variational bound, and RedDiff which employs descending time from $T$ to 0. Red-Diff formulates the loss function at each time step $t$ as follows:

$$||y - f(\mu)||^2 + w_t(sg[\epsilon - \epsilon_\theta(z_t, t)])^T\mu \quad (10)$$

where $w_t$ is a loss weighing term adjusted by Red-Diff and $sg$ denotes a stop-gradient operator used by Red-Diff to prevent optimization instability, which may arise due to the denoising function's lack of smoothness at low-noise levels (Yang et al., 2024).

Because Red-Diff employs a simple variational posterior that directly optimizes an image at the noise free ($t = 0$) level only, it is inherently incapable to capture uncertainty in $x$ due to its mode-seeking behaviour. Additionally, its optimization process is *biased* because it relies on annealing time during the diffusion process rather than randomly sampling time points. We demonstrate that in contrast, our VIPaint framework better models posterior uncertainty, enables stable optimization of an unbiased variational bound, and can be applied to both pixel-based and latent DMs.

## 4   VIPAINT: VARIATIONAL INFERENCE OF DIFFUSIONS FOR INPAINTING

Given a pre-trained diffusion model, our framework defines a structured family of distributions on the latent variable $z$, which is optimized for each partial observation. This simple and general approach can be applied both to models which diffuse in the pixel space, or latent space.

Rather than optimizing a posterior or latent samples at a single noise level and annealing through noise levels, our *VIPaint* method constructs a joint posterior that is hierarchical across a subset of $z_t$, introducing a hierarchical structure to our inference. The parameters $\lambda$ of this posterior are optimized separately for each query to steer our proposed variational distribution to be semantically aligned with the observation, while searching for consistent images within the prior higher ranged ($[T_e, T]$) latent space $z$. Once fit, the second stage involves sampling from this posterior and utilizes DPS gradient updates in the low ranged ($[0, T_s]$) latent space, see...

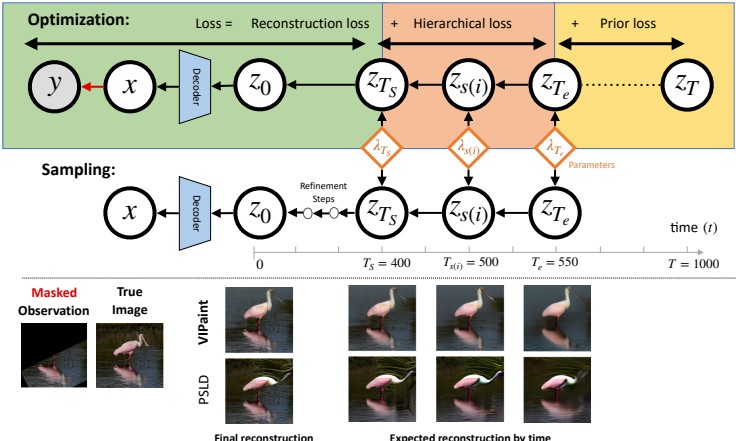

Figure 3: *Top:* The hierarchical approximate posterior of VIPaint is defined over a coarse sequence of intermediate latent steps between $T_e$ and $T_s$. During optimization, the variational parameters $\lambda$ defining the posterior on a subset of latent times are fit via a prior loss on times above $T_e$, a hierarchical loss defined across $K$ intermediate times, and a reconstruction loss estimated using a one-step approximation $p_\theta(x|z_{T_s})$ from the posterior samples. *Bottom:* After variational inference, samples from the hierarchical posterior (now *aligned* with the observation) transition smoothly in the intermediate latent space $[0, T_s]$ via gradient updates. Note that samples at $T_e$ and $T_s$ are aligned much better for VIPaint then the baseline PSLD (Rout et al., 2023), whose predications at $T_e = 550$ contain artifacts which subsequent steps cannot correct.

## 4.1 DEFINING THE VARIATIONAL POSTERIOR

Our variational posterior is defined on the latent space $z$, at multiple noise levels, to capture global semantics in the observation $y$. Because diffusion models encode a rich, multi-scale representation in the latent space $z$, we hypothesize that a range of timesteps in between contain critical relevant information, that we aim to capture through our posterior. We avoid having our posterior be (explicitly) defined on timesteps $(T_e, T]$ which behaves close to Gaussian noise, and $[0, T_s)$ which contains only fine-details, and define a hierarchical posterior over $K$ intermediate timesteps.

VIPaint retains the non-linearity and complexity of the noise prediction model $\theta$ and follows the sample generating reverse diffusion process to produce inpaintings $x$. The variational parameters $\lambda$ stochastically bias this sample generation towards samples from the true posterior induced by observation $y$, and can be factorized as:

$$q(x) = \int_z q(x \mid z_{T_s}) \left( \prod_{i=1}^{K-1} q_\lambda(z_{s(i)} \mid z_{s(i+1)}) \right) q_\lambda(z_{T_e}) \, dz, \qquad (11)$$

where timesteps $(T_s, T_e)$ define the boundaries of our variational posterior along the diffusion timesteps. We model $q(x \mid z_{T_s}) = \int_{z_0} p(x \mid z_0)p(z_0 \mid z_{T_s}) \, dz_0$, where $p(x \mid z_0)$ is a factorized Gaussian likelihood for pixel-based diffusion models, or a decoder for LDMs. $p(z_0|z_{T_s})$ also follows the prior with a one-step expected mean prediction $\mathbb{E}[\hat{z}_0|z_{T_s}]$ as in Eq. (3) and negligible standard deviation. For our highest timestep $T_e$, we let our posterior $q_\lambda(z_{T_e})$ be a simple Gaussian $\mathcal{N}(\mu_{T_e}, \tau_{T_e})$ with variational parameters $(\mu_{T_e}, \tau_{T_e})$ defined over each pixel in the image or its encoding. Denoting $s(i)$ as the timestep preceding $s(i + 1)$ for all $i \in [1, K - 1]$, and generalizing the hierarchical VAE approximation of Agarwal et al. (2023), we let our conditional equal

$$q_\lambda(z_{s(i)} \mid z_{s(i+1)}) = \mathcal{N}(z_{s(i)} \mid \gamma_{s(i)}\hat{z}_{s(i)} + (1 - \gamma_{s(i)})\mu_{s(i)}, \tau_{s(i)}^2), \qquad (12)$$

where $\hat{z}_{s(i)} = \hat{z}_{s(i)}(\theta, z_t, t)$ is the mean prediction of the prior diffusion model $p(z_{(s(i))}|z_{(s(i+1))})$, and $\lambda = \{\mu_{T_e}, \tau_{T_e}, (\gamma_{s(i)}, \mu_{s(i)}, \tau_{s(i)})_{i=1}^{K-1}\}$ are the set of variational parameters. We use $y$ to initialize $\mu_{s(i)}$ by first encoding it using the encoder and then scaling it by the forward diffusion parameter $\alpha_{s(i)}$. We also use the prior $(\sigma_t)$ and posterior $(\tilde{\sigma}_t)$ from the diffusion noise schedule to initialize our posterior variance, Appendix E.1 for details.

At every timestep $i$, the mean of the posterior interpolates between the noise prediction network $\hat{z}_{s(i)}$ and a contextual parameter $\mu_{s(i)}$ for a given query $y$. This is key when reusing the diffusion

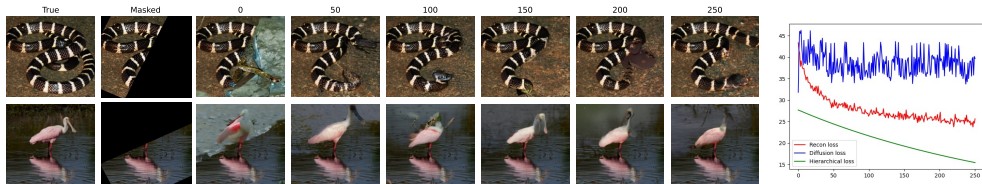

Figure 4: We show the progress of fitting VIPaint's posterior and draw samples after every 50 iterations of inference for two test cases. We see that VIPaint quickly figures out the semantics in the underlying image within 50 optimization iterations.

prior to adjust the posterior to align precisely with a particular observation $y$, without the need to re-train $\theta$. Previous work (Song et al., 2021b; Lugmayr et al., 2022; Kawar et al., 2022; Song et al., 2024) uses linear combinations between the observed $y$ and generated sample $z_t$, but either use hard constraints or fixed weights that are manually tuned. Instead, we incorporate free latent parameters and optimize them using the variational bound derived below.

## 4.2 PHASE 1 : OPTIMIZATION

To fit our hierarchical posterior, we optimize the variational lower bound (VLB) of the marginal likelihood of the observation $y$ (we derive this in Appendix C): $-\log p(y) \leq$

$$\underbrace{-\mathbb{E}_q[\log p_\theta(y|z_{T_s})]}_{\text{reconstruction loss}} + \beta \underbrace{\sum_{i=1}^{K-1} D\Big[q_\lambda(z_{s(i)}|z_{s(i+1)})||p_\theta(z_{s(i)}|z_{s(i+1)}))\Big]}_{\text{hierarchical loss}} + \beta \underbrace{\mathcal{L}_{(T_e,T)}(z_{T_e})}_{\text{diffusion loss}}, \quad (13)$$

where VIPaint seeks latent-code distributions that assign high likelihood to the observed features $y$, while simultaneously aligning with the medium-to-high noise levels encoding image semantics via weight $\beta > 1$ (Higgins et al., 2017; Agarwal et al., 2023).

**Diffusion Loss.** $\mathcal{L}_{(T_e,T)}(z_{T_e})$ is essentially a restriction of Eq. (5) to a small set of times $(T_e, T)$ with high noise levels. This diffusion term queries the latent space of the diffusion model at high noise levels ($> T_e$) to guide the posterior $q(z_{T_e})$ towards a distribution in the prior latent space to be consistent with the observation $y$ in high-level semantics. Following prior work, instead of summing this loss over all $t > T_e$, we sample timesteps $t \sim \mathcal{U}(T_e, T)$ defined on a non-uniform discretization (Karras et al., 2022), yielding an unbiased estimate of the loss as (see App. C) :

$$\mathcal{L}_{(T_e,T)}(z_{T_e}) = \frac{T - T_e}{2}\mathbb{E}_{t\sim\mathcal{U}(T_e,T),q(z_t|z_{T_e})}D[q(z_{t-1}|z_t, z_{T_e})||p_\theta(z_{t-1}|z_t)]. \quad (14)$$

**Hierarchical Loss.** For subsequent steps in our Markov posterior, the hierarchical loss closes the gap between our posterior $q(z_{s(i)}|z_{s(i+1)})$ and the prior $p(z_{s(i)}|z_{s(i+1)})$ at each step $i$ by minimizing the KL divergence (an analytic function of the means and variances).

**Reconstruction Loss.** While the posterior aligns with the prior latent space, the reconstruction term guides the samples from the posterior $z_{T_s}$ to be closer to the observations $y$. We utilize Tweedie's formula to approximate $z_0$ and then, for latent diffusion models, we use decoder up-sampling to produce image $\hat{x}$. We follow the L1 reconstruction loss that was used to pre-train the diffusion models. For latent diffusion models specifically for the task of image inpainting, we add the perceptual loss (Zhang et al., 2018) that was also originally used to train the decoder. Fig. 9 (Appendix) shows an ablation that adding such a term helps avoid blurry reconstructions.

All the loss terms in Eq. (13) are stochastically and differentiably estimated based on samples from the hierarchical posterior, enabling joint optimization. From Eq. 13, if the posterior is only defined on the noise-free level $z_0$ as in Red-Diff (Mardani et al., 2023), the VIPaint objective reduces to an objective mentioned in their work. However, VIPaint strategically avoids low noise levels in its posterior and decreases training instabilities as observed by RedDiff.

## 4.3 PHASE 2 : SAMPLING

After optimization, samples $z_{T_s}$ are drawn from $\prod_{i=1}^{K} q_\lambda(z_{s(i-1)}|z_{s(i)})q_\lambda(z_{T_e})$, that is now seman-tically aligned with the observation, using ancestral sampling on our $K$ level hierarchical posterior starting from $T_e$ to $T_s$. This step gradually adds more semantic details in samples. Additionally, VIPaint utilizes DPS gradient updates to iteratively refine $z_{T_s}$ to produce $z_0$ to ensure fine-grained consistency with $y$, as this approximation is effective in low-noise regimes. See Fig. 3.

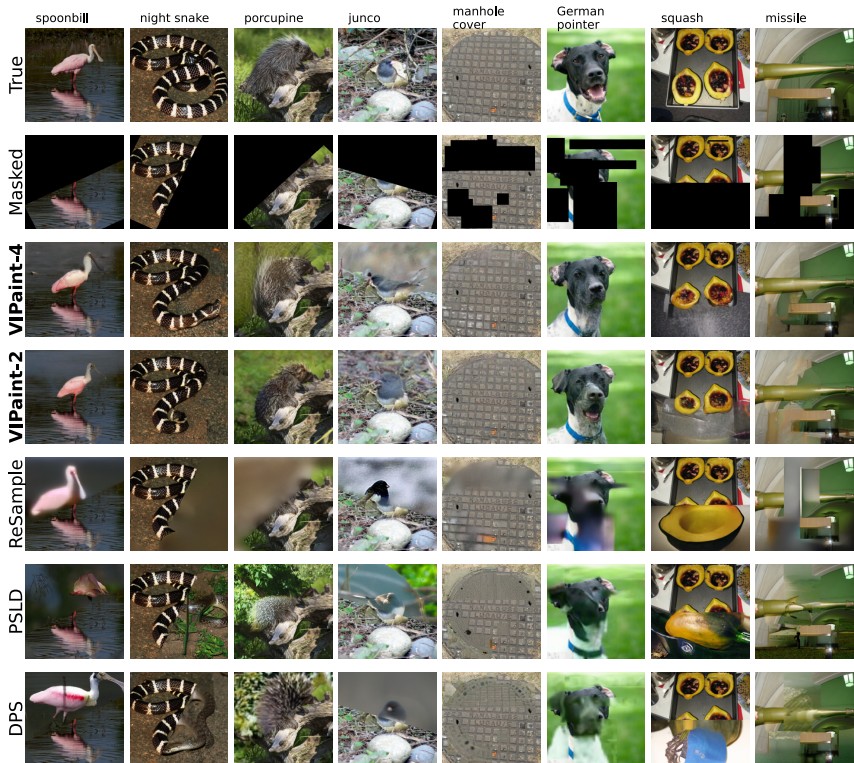

Figure 5: Image completion results on Imagenet256 using the LDM prior for Rotated Window and Random Masking schemes shown in the second row. We show an inpainting from each method in the following four rows. DPS, PSLD, and ReSample show blurry inpaintings of widely varying quality. In contrast, VIPaint interprets the global semantics in the observed image and produces *very* realistic images. Please find more qualitative plots for LSUN-church in the Appendix Fig. 15.

## 5 EXPERIMENTS & RESULTS

### 5.1 EXPERIMENTAL SETUP

| Task | VIPaint-4 | VIPaint-2 | CoPaint-TT | CoPaint | RePaint | DPS | Blended | RedDiff | RedDiff-V |
|---|---|---|---|---|---|---|---|---|---|
| Rotated Window | **0.289** | 0.300 | 0.316 | 0.347 | 0.3213 | 0.3203 | 0.3409 | 0.463 | 0.407 |
| Random Mask | 0.231 | **0.227** | 0.245 | 0.278 | 0.2575 | 0.2880 | 0.2763 | 0.409 | 0.671 |

| Task | Imagenet-256 | | | | | LSUN-Church | | | |
|---|---|---|---|---|---|---|---|---|---|
| | VIPaint-4 | VIPaint-2 | ReSample | PSLD | DPS | VIPaint-2 | ReSample | PSLD | DPS |
| Rotated Window | **0.358** | 0.392 | 0.537 | 0.576 | 0.606 | **0.455** | 0.510 | 0.541 | 0.502 |
| Random Mask | **0.373** | 0.409 | 0.559 | 0.583 | 0.607 | **0.439** | 0.485 | 0.523 | 0.490 |
| Small Mask | 0.292 | **0.197** | 0.381 | 0.534 | 0.564 | **0.299** | 0.374 | 0.413 | 0.421 |

Table 1: Quantitative results (LPIPS, lower is better) for ImageNet64 for the task of image inpainting using pixel-based EDM prior (*top*) and Imagenet-256 and LSUN-Church using LDM priors (*bottom*). LPIPS is estimated as the mean score of 10 inpaintings with respect to the true image, averaged across the test set. VIPaint has superior performance (highlighted in **bold**) in nearly all cases. We underline the second best method. Fig. 11 in the appendix has further comparisons.

We conduct experiments across 3 image datasets: LSUN-Church Yu et al. (2015), ImageNet-64 and ImageNet-256 Deng et al. (2009). For ImageNet-64, we use the class-conditioned pixel-space "EDM" diffusion model Karras et al. (2022) with the pre-trained score network provided by the authors. For LSUN-Churches256 and ImageNet256 we use the pre-trained latent diffusion models from Rombach et al. (2022b). Then, we sample 100 non-cherry-picked test images across the three datasets. We consider three masking patterns: 1) a small mask distribution (Zhao et al., 2021) that

Figure 6: Image completion results on ImageNet64 using a conditional pixel-based EDM prior for image inpainting (Random Masking and Rotated Window schemes) shown in the second row. We show an inpainting for each method in the following rows. Even though the prior diffusion model for ImageNet is conditioned on class labels, inpaintings for baseline methods are inconsistent with the observed image. RePaint and CoPaint is typically more accurate than other baselines, but still produce incoherent samples unless masks are small. In contrast, VIPaint interprets the global semantics of the observed image while enforcing consistency with the few observed pixels.

masks up to 80% of each image 1) a random mask distribution that uses a similar setup masking between 40 and 80% of each image, and 2) a randomly rotated masking window that masks at least half of the image. By masking large portions of each image we ensure a sufficiently challenging benchmark for in-painting. Some prior works (Chung et al. (2023)) only employ masks covering a small percentage of pixels. Recent works (RePaint Lugmayr et al. (2022), CoPaint Zhang et al. (2023)) note that these methods struggle with larger masks. For each test image, we evaluate each method across 10 realizations per test image, totalling 1000 inpaintings. We test VIPaint for other linear inverse problems like super-resolution and gaussian deblurring in Appendix H.2

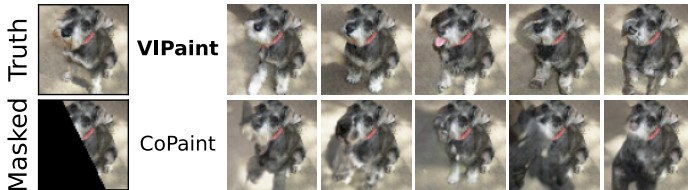

Figure 7: Sample completions comparing VIPaint with the best performing baseline, CoPaint, for a test image. We show the true and masked images, and 5 in-painted samples for each method. For an extended comparison see Appendix Fig. 22. CoPaint shows high variance in the quality of image completions, while VIPaint yields coherent samples while capturing uncertainty.

**Comparison.** We compare VIPaint with several recent methods that directly apply the diffusion models trained in the pixel space: *i)* blending methods: *blended* (Song et al., 2021b) and *RePaint* Lugmayr et al. (2022) ; *ii)* Sampling methods: *DPS* (Chung et al., 2023), and *CoPaint* (Zhang et al., 2023) and *iii)* variational approximations: *RED-Diff* Mardani et al. (2023). Although not exhaustive, this set of methods summarizes recent developments in the state-of-the-art for image inpainting. For latent diffusion models, we compare VIPaint with *DPS*, *PSLD* Rout et al. (2023) and ReSample Song et al. (2024) which are state-of-the-art for inpainting with latent diffusion models. Please see Appendix E.2 for additional details on their implementation. Since large masks in images can induce high-uncertainty in the image, Peak-Signal-To-Noise-Ratio (PSNR) is not very well defined for this task. While metrics like Kernel Inception Distance Bińkowski et al. (2018) require a large set of images, we report the Learned Perceptual Image Patch Similarity (LPIPS) Zhang et al. (2018) metric in Table 1. We report Peak-Signal-To-Noise-Ratio (PSNR) for some other linear inverse problems like Super Resolution and Gaussian Deblurring in Table 4 (Appendix). We show qualitative images across methods for ImageNet64 in Fig 6, ImageNet256 in Fig. 5 and LSUN-Church in Fig. 15 (Appendix). For tasks like super-resolution and Gaussian Deblurring, we show qualitative results in Fig. 12, 13 and 14. Additionally, we visualize multiple inpaintings in Fig. 7.

**Hyperparameters.** We use the notation VIPaint-$K$ to denote the number of steps in the hierarchical posterior in our experiments. We found empirically that discretizations and hyperparameters of VIPaint translate well between models using the same noise schedule (as shown for the LSUN and ImageNet-256 latent diffusion models). Please see Appendix E.1 for more details.

## 5.2 RESULTS

**VIPaint enforces consistency with large masking ratios.** Table 1 reports LPIPS scores for the task of image inpainting with large masking ratios using pixel and latent-based diffusion models, respectively. For pixel-based diffusion models, we see that RED-Diff and DPS perform poorly. RePaint, CoPaint and CoPaint-TT show relatively better scores, but do not match VIPaint across any dataset or masking pattern. We show imputations for multiple test examples in Fig. 6, 5 and 15 (Appendix) to highlight differences in inference methods. We see that VIPaint consistently produces plausible inpaintings while other methods fail to complete images for larger masking ratios meaningfully.

**VIPaint yields multiple plausible reconstructions in the case of high uncertainty.** We compare VIPaint with the best performing baseline, CoPaint across multiple sample inpaintings in Fig. 7, a more comprehensive comparison is in Appendix (Fig 18-24). We observe that VIPaint produces multiple visually-plausible imputations while not violating the consistency across observations. We show diversity in possible imputations using different class conditioning using VIPaint in Fig. 24.

**VIPaint smoothly trades off time and sample quality.** VIPaint-2, utilizing a two-step hierarchy naturally is the fastest choice for any $k$ in VIPaint-$K$. It is comparable with other baselines with respect to time (for a more detailed analysis, please refer to Appendix F). However, from Tables 1, we see a remarkable gain in performance when compared with other baselines. VIPaint-4 converges a bit more slowly (Fig. 10), but ultimately reaches the best solutions.

## 6 CONCLUSION

We present VIPaint, a simple and a general approach to adapt diffusion models for image inpainting and other inverse problems. We take widely used (latent) diffusion generative models, allocate variational parameters for the latent codes of each partial observation, and fit the parameters stochastically to optimize the induced variational bound. The simple but flexible structure of our bounds allows VIPaint to outperform previous sampling and variational methods when uncertainty is high.

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
