# A APPENDIX

# B DIFFUSION MODELS : DEFINITION & TRAINING PROCEDURE RECAP

## B.1 FORWARD TIME DIFFUSION PROCESS

The background and expressions on forward diffusion process is taken from Kingma et al. (2021b) and included here for completeness. Re-iterating Eq. 1, we have the forward diffusion as:

$$q(z_t \mid x) = \mathcal{N}(\alpha_t x, \sigma_t^2 I). \tag{15}$$

**Forward Conditional** $q(z_t|z_s)$**:** The distribution $q(z_t|z_s)$ for any $t > s$ are also Gaussian, and from Kingma et al. (2021b), we can re-write as

$$\mathcal{N}(\alpha_{t|s} z_s, \sigma_{t|s}^2 I) \tag{16}$$

$$\text{where,} \quad \alpha_{t|s} = \alpha_t/\alpha_s, \tag{17}$$

$$\text{and,} \quad \sigma_{t|s}^2 = \sigma_t^2 - \alpha_{t|s}^2 \sigma_s^2 \tag{18}$$

**Reverse Conditional,** $q(z_s|z_t, x)$**:** The posterior $q(z_s|z_t, x)$ from Kingma et al. (2021b) can be written as:

$$q(z_s|z_t, x) = \mathcal{N}(\mu_Q(z_t, x; s, t), \sigma_Q^2(s, t)I) \tag{19}$$

$$\text{where,} \quad \sigma_Q^2(s, t) = \sigma_{t|s}^2 \sigma_s^2 / \sigma_t^2 \tag{20}$$

$$\text{and,} \quad \mu_Q(z_t, x; s, t) = \frac{\alpha_{t|s} \sigma_s^2}{\sigma_t^2} z_t + \frac{\alpha_s \sigma_{t|s}^2}{\sigma_t^2} x \tag{21}$$

## B.2 REVERSE DIFFUSION : DEFINING $p_\theta(z_s|z_t)$

Here, we describe in detail the conditional reverse model distributions $p_\theta(z_s|z_t)$ for the two cases of variance-exploding and variance preserving diffusion process. Given these formulations, it is straightforward to compute the KL distance between our posterior $q_\lambda(z_s|z_t, y)$ and the prior $p_\theta(z_s|z_t)$ in our loss objective (Eq. 13) since both are conditionally Gausian distributions and computing the KL between two Gaussians can be done in closed form.

**Variance Exploding Diffusion Process** In this case, $\alpha_t = 1$ and $\sigma_t$ is usually in the range $[0.002, 50]$ Song et al. (2021b). We follow the ancestral sampling rule from the same work to define our prior conditional Gaussian distributions $p_\theta(z_s|z_t)$ :

$$p_\theta(z_s|z_t) = \mathcal{N}(\mu_\theta(z_t; s, t), \sigma_Q^2(s, t)I) \tag{22}$$

$$\text{where,} \quad \sigma_Q^2(s, t) = (\sigma_t^2 - \sigma_s^2)\frac{\sigma_s^2}{\sigma_t^2} \tag{23}$$

$$\text{and,} \quad \mu_\theta(z_t; s, t) = \frac{\sigma_s^2}{\sigma_t^2} z_t + \frac{\sigma_t^2 - \sigma_s^2}{\sigma_t^2} \hat{x}_\theta(z_t, t) \tag{24}$$

where $\hat{x}_\theta(z_t, t) = z_t - \sqrt{(\sigma_t^2 - \sigma_s^2)} * \epsilon_\theta(z_t, t)$

**Variance Preserving Diffusion Process** In this case, $\alpha_t = \sqrt{1 - \sigma_t^2}$ and $\sigma_t^2$ is usually in the range $[0.001, 1]$ Ho et al. (2020b). We follow the DDIM sampling rule Song et al. (2021a) to define our prior conditional Gaussian distributions $p_\theta(z_s|z_t)$. This sampling rule is widely used to generate unconditional samples in small number of steps, and naturally becomes a key design choice of our prior. Here,

$$p_\theta(z_s|z_t) = \mathcal{N}(\mu_\theta(z_t; s, t), \sigma_Q^2(s, t)I) \tag{25}$$

$$\text{where,} \quad \sigma_Q^2(s, t) = \eta(\frac{1 - \alpha_{t-1}}{1 - \alpha_t})(1 - \frac{\alpha_t}{\alpha_{t-1}}) \tag{26}$$

$$\text{and,} \quad \mu_\theta(z_t; s, t) = \sqrt{\alpha_{t-1}}\hat{x}_\theta(z_t, t) + \sqrt{1 - \alpha_t - \sigma_t^2}\epsilon_\theta(z_t, t) \tag{27}$$

where $\hat{x}_\theta(z_t, t) = \frac{z_t - \sqrt{1 - \alpha_t}\epsilon_\theta(z_t, t)}{\sqrt{\alpha_t}}$. This schedule is adopted by the Latent Diffusion models.

## B.3 DERIVATION OF OBJECTIVE FOR TRAINING DIFFUSION MODELS: $\mathcal{L}_{(0,T)}(z_0)$

The usual variational bound on the negative loglikehood on data $x$: $\mathbb{E}[-\log p_\theta(x)] \leq \mathbb{E}_q[-\log \frac{p_\theta(z_{0:T})}{q(z_{1:T}|z_0,x)}] = \mathbb{E}_q[-\log p(z_T) - \sum_{t=1}^T \log \frac{p_\theta(z_{t-1}|z_t)}{q(z_t|z_{t-1})}]$. Let $0 < s < t < T$, we expand this derivation from Ho et al. (2020b) as follows:

$$\mathcal{L} = \mathbb{E}_q \left[ \log \frac{q(z_{1:T}|z_0)}{p_\theta(z_0)} \right] \tag{28}$$

$$= \mathbb{E}_q \left[ -\log p(z_T) + \sum_{t \geq 1} \log \frac{q(z_t|z_s)}{p_\theta(z_s|z_t)} \right] \tag{29}$$

$$= \mathbb{E}_q \left[ -\log p(z_T) + \sum_{t > 1} \log \frac{q(z_t|z_s)}{p_\theta(z_s|z_t)} + \log \frac{q(z_1|z_0)}{p_\theta(z_0|z_1)} \right] \tag{30}$$

$$= \mathbb{E}_q \left[ -\log p(z_T) + \sum_{t > 1} \log \frac{q(z_s|z_t, z_0)}{p_\theta(z_s|z_t)} \cdot \frac{q(z_t|z_0)}{q(z_s|z_0)} + \log \frac{q(z_1|z_0)}{p_\theta(z_0|z_1)} \right] \tag{31}$$

$$= \mathbb{E}_q \left[ -\log \frac{p(z_T)}{q(z_T|z_0)} + \underbrace{\sum_{t > 1} \log \frac{q(z_s|z_t, z_0)}{p_\theta(z_s|z_t)}}_{\text{diffusion loss } \mathcal{L}_{(0,T)}(z_0)} - \log p_\theta(z_0|z_1) \right] \tag{32}$$

## C  VIPAINT : VI METHOD USING DIFFUSION MODELS AS PRIORS

### C.1  DERIVATION OF VIPAINT'S TRAINING OBJECTIVE

As specified in the main paper, we define a variational distribution over the latent space variable $z$ as $q_\lambda(z)$ and re-use the diffusion prior to generate $x \sim p_\theta(x \mid z)$. We derive the variational objective here:

$$
\begin{aligned}
&\mathcal{L}_N(\lambda; y) \\
&= \mathbb{E}_{q_\lambda(z,x)}[\log p_\theta(y, x, z) - \log q_\lambda(z, x \mid y)] \\
&= \mathbb{E}_{q_\lambda(z,x)}[\log p_\theta(z) + \log p_\theta(x \mid z_{T_s}) + \log p_\theta(y \mid z_{T_s}) - \log q_\lambda(z) - \log q_\lambda(x \mid z_{T_s})] \\
&= \mathbb{E}_{q_\lambda(z)}[\log p_\theta(y \mid z_{T_s}) + \log p_\theta(z) - \log q_\lambda(z)] \\
&= \mathbb{E}_{q_\lambda(z)}[\log p_\theta(y \mid z_{T_s})] - \mathbb{E}_{q_\lambda(z)}[\log q_\lambda(z) - \log p_\theta(z)] \\
&= \mathbb{E}_{q_\lambda(z)}[\log p_\theta(y \mid z_{T_s})] - \mathbb{E}_{q_\lambda(z)}[\log q_\lambda(z) - \log p_\theta(z)] \\
&= \mathbb{E}_{q_\lambda(z)}[\log p_\theta(y \mid z_{T_s})] - \mathbb{E}_{q_\lambda(z)}[\sum_{i=0}^{K-1} \log q_\lambda(z_{s(i)} \mid z_{s(i+1)}) + \log q_\lambda(z_{T_e}) - \sum_{i=0}^{K-1} \log p_\theta(z_{s(i)} \mid z_{s(i+1)}) - \log p_\theta(z_{T_e})] \\
&= \mathbb{E}_{q_\lambda(z)}[\log p_\theta(y \mid z_{T_s})] - \sum_{i=0}^{K-1} D[q_\lambda(z_{s(i)} \mid z_{s(i+1)}) || p_\theta(z_{s(i)} \mid z_{s(i+1)})] - \underbrace{D(q(z_{T_e}) || p(z_{z_{T_e}}))}_{\mathcal{L}_{(T_e, T)}(z_{T_e})}
\end{aligned}
\tag{33}
$$

Negating the above objective, we get Eq. 13 in the main paper. Now, let's derive the third term $L_{(T_e, T)}(z_{T_e})$ following section B.3

### C.2  DERIVATION OF $L_{(T_e, T)}(z_{T_e})$

For any $T_e < s < t < T$, we have :

$$
\mathbb{E}_{z_{T_e} \sim q_\lambda(z_{T_e})} \left[ \log \frac{q(z_{T_e+1:T} | z_{T_e})}{p_\theta(z_{T_e:T})} \right]
\tag{34}
$$

$$
= \mathbb{E}_{z_{T_e} \sim q_\lambda(z_{T_e})} \left[ - \log p(z_T) + \sum_{t \geq T_e} \log \frac{q(z_t | z_s)}{p_\theta(z_s | z_t)} \right]
\tag{35}
$$

$$
= \mathbb{E}_{z_{T_e} \sim q_\lambda(z_{T_e})} \left[ - \log p(z_T) + \sum_{t > T_e} \log \frac{q(z_t | z_s)}{p_\theta(z_s | z_t)} + \log \frac{q(z_{T_e+1} | z_{T_e})}{p_\theta(z_{T_e} | z_{T_e+1})} \right]
\tag{36}
$$

$$
= \mathbb{E}_{z_{T_e} \sim q_\lambda(z_{T_e})} \left[ - \log p(z_T) + \sum_{t > T_e} \log \frac{q(z_s | z_t, z_{T_e})}{p_\theta(z_s | z_t)} \cdot \frac{q(z_t | z_{T_e})}{q(z_s | z_{T_e})} + \log \frac{q(z_{T_e+1} | z_{T_e})}{p_\theta(z_{T_e} | z_{T_e+1})} \right]
\tag{37}
$$

$$
= \mathbb{E}_{z_{T_e} \sim q_\lambda(z_{T_e})} \left[ - \log \frac{p(z_T)}{q(z_T | z_{T_e})} + \underbrace{\sum_{t > T_e} \log \frac{q(z_s | z_t, z_{T_e})}{p_\theta(z_s | z_t)}}_{\text{diffusion loss } \mathcal{L}_{(T_e, T)}(z_{T_e})} - \log p_\theta(z_{T_e} | z_{T_e+1}) \right]
\tag{38}
$$

The first and third term can be stochastically and differentially estimated using standard techniques. Following Kingma et al. (2021b), we derive an estimator for the diffusion loss $\mathcal{L}_{(T_e, T)}(z_{T_e})$. In the case of finite timesteps $t > T_e$, this loss is:

$$
\mathcal{L}_{(T_e, T)}(z_{T_e}) = \sum_{t > T_e} \mathbb{E}_{q(z_t | z_{T_e})} D[q(z_s | z_t, z_{T_e}) || p_\theta(z_s | z_t)]
\tag{39}
$$

**Estimator of** $\mathcal{L}_{(T_e,T)}(z_{T_e})$   Reparametering $z_t \sim q(z_t|z_{T_e})$ as $z_t = \alpha_{t|T_e} z_{T_e} + \sigma_{t|T_e} \epsilon$, where $\epsilon \sim \mathcal{N}(0,1)$, and to avoid having to compute all $T - T_e$ terms when calculating the diffusion loss, we construct an unbiased estimator of $\mathcal{L}_{(T_e,T)}(z_{T_e})$ using

$$\mathcal{L}_{(T_e,T)}(z_{T_e}) = \frac{T-T_e}{2} \mathbb{E}_{\epsilon,t\sim\mathcal{U}(T_e,T)}[D(q(z_s|z_t,z_{T_e})||p_\theta(z_s|z_t))] \tag{40}$$

where $\mathcal{U}(T_e,T)$ is a uniform distribution to sample $T_e < t \leq T$ from a non-uniform descretization of timesteps using Karras et al. (2022).

Now, we elaborate on the expression $q(z_s|z_t, z_{T_e})$ and $p(z_s|z_t)$ for any $T_e < s < t < T$.

### C.2.1   $q(z_s|z_t, z_{T_e})$

Our posterior at $T_e$ is $q(z_{T_e}) = \mathcal{N}(\mu_{T_e}, \tau_{T_e}^2)$. For any $T_e < s < t < T$, we have $q(z_s|z_{T_e}) = \mathcal{N}(\alpha_{s|T_e} z_{T_e}, \tau_{s|T_e}^2)$ and $q(z_t|z_s) = \mathcal{N}(\alpha_{t|s} z_s, \sigma_{t|s}^2)$, yielding the posterior :

$$q(z_s|z_t, z_{T_e}) = \mathcal{N}(\mu_Q(z_t, z_{T_e}; s, t, T_e), \sigma_Q^2(s, t, T_e)I) \tag{41}$$

$$\text{where,} \quad \sigma_Q^2(s, t, T_e) = \sigma_{t|s}^2 \frac{\tau_{s|T_e}^2}{\sigma_{t|s}^2 + \alpha_{s|T_e}^2 \tau_{s|T_e}^2} \tag{42}$$

$$\text{and,} \quad \mu_Q(z_t, z_{T_e}; s, t, T_e) = \sigma_Q^2 \left( \frac{\alpha_{s|T_e}}{\tau_{s|T_e}^2} z_{T_e} + \frac{\alpha_{t|s}}{\sigma_{t|s}^2} z_t \right) \tag{43}$$

$$= \frac{\alpha_{s|T_e} \sigma_{t|s}^2}{(\sigma_{t|s}^2 + \alpha_{s|T_e}^2 \tau_{s|T_e}^2)} z_{T_e} + \frac{\alpha_{t|s} \tau_{s|T_e}^2}{(\sigma_{t|s}^2 + \alpha_{s|T_e}^2 \tau_{s|T_e}^2)} z_t \tag{44}$$

### C.2.2   $p(z_s|z_t)$

The conditional model distributions can be chosen as:

$$p_\theta(z_s|z_t) = q(z_s|z_t, z_{T_e} = \hat{z}_{\theta,T_e}(z_t,t)) = \mathcal{N}(z_s; \mu_\theta(z_t, z_{T_e}; s, t, T_e), \sigma_Q^2(s, t, T_e)) \tag{45}$$

$$\text{where,} \quad \mu_\theta(z_t, z_{T_e}; s, t, T_e) = \frac{\alpha_{s|T_e} \sigma_{t|s}^2}{(\sigma_{t|s}^2 + \alpha_{s|T_e}^2 \tau_{s|T_e}^2)} \hat{z}_{\theta,T_e}(z_t,t) + \frac{\alpha_{t|s} \tau_{s|T_e}^2}{(\sigma_{t|s}^2 + \alpha_{s|T_e}^2 \tau_{s|T_e}^2)} z_t \tag{46}$$

$$\text{and,} \quad \sigma_Q^2(s, t, T_e) = \sigma_{t|s}^2 \frac{\sigma_{s|T_e}^2}{\sigma_{t|s}^2 + \alpha_{s|T_e}^2 \sigma_{s|T_e}^2} \tag{47}$$

where $\hat{z}_{\theta,T_e}(z_t,t) = \frac{z_t - \sigma_{t|T_e} * \epsilon_\theta(z_t,t)}{\alpha_{t|T_e}}$

### C.3   TIME ANALYSIS

For a $K$ step hierarchical posterior, each optimization step to fit the posterior requires $K + 1$ number of function evaluations (#NFEs) of the denoising network, where it uses $K$ #NFEs during sampling from the posterior and 1 to compute the diffusion prior loss. Gradients are back-propagated through the denoising network during optimization, making each optimization step more informative than other works iteratively refining samples. We see the progress of fitting VIPaint's posterior in Fig. 4 and often times, we observe that convergence occurs as low as in 50 optimization steps. Thus, VIPaint with 2 steps in its hierarchy can effectively infer the semantics in the image using only a cumulative of $50 * (2 + 1) = 150$ NFEs of the denoisinng network. The optimization time is $O(K)$, where $K << T$. but also linearly increases the inference time.

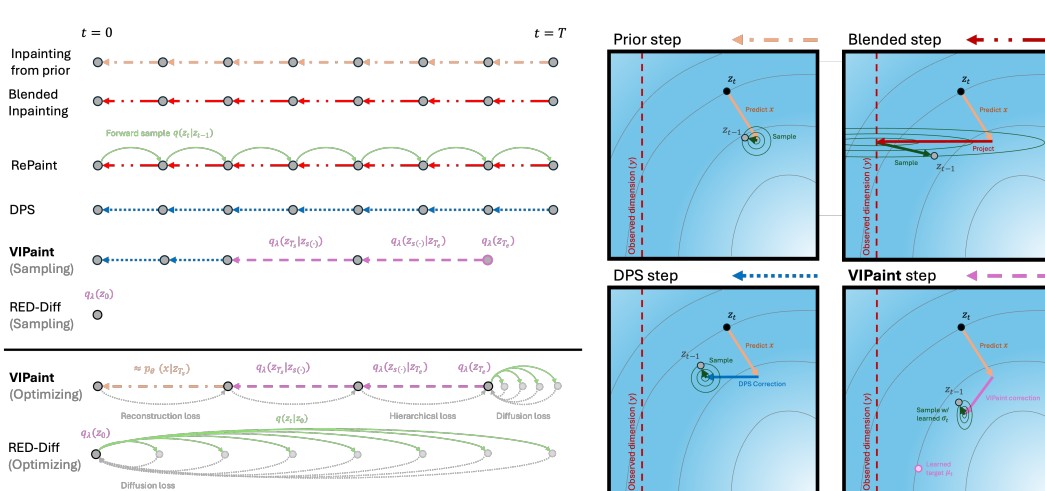

Figure 8: Expanded comparison of methods for diffusion model-based inpainting. **Left:** Timeline illustration of sampling steps with time flowing rightward from $t = 0$ (clean images) to $t = T$ (pure noise). Orange arrows indicate a single step of ancestral sampling under the generative prior $p_\theta(z_{t-1}|z_t)$. Red arrows indicate a single step of the *Blended* approximation of $p_\theta(z_{t-1}|z_t, y)$, while blue arrows indicate a single step of the *DPS* approximation. Green arrows indicate steps forward in time according to the diffusion process $q(z_t|z_{<t})$. Methods such as *RePaint* and *CoPaint* alternate between forward and reverse steps. Purple arrows indicate sampling from a step in the hierarchical **VIPaint** posterior $q_\lambda(z_{s(i-1)}|z_{s(i)})$. Both **VIPaint** and *RED-Diff* (without annealing) involve an initial optimization stage to fit variational parameters per-image. Gray arrows indicate the flow of gradient information during this optimization stage. Gray points are steps only used during optimization. **Right:** Illustration of each reverse-time sampling step in 2 dimensions. The horizontal dimension is assumed to be observed at the value marked by the red line. Each approach begins by computing $p_\theta(z_{t-1}|z_t)$ via a prediction of $x$ using the pre-trained denoising network $\hat{x}_\theta(z_t, t)$. *Blended* replaces observed dimensions with $q(z_{t-1}|y)$. *DPS* updates $p_\theta(z_{t-1}|z_t)$ according to a single-step approximation to the likelihood $p_\theta(y|z_{t-1})$. Finally, **VIPaint**, uses a learned variational distribution $q_\lambda(z_{t-1}|z_t)$, which can be seen as interpolating between the prediction of $x$ and a variational parameter $\mu_t$, coupled with a learned variance.

## D    EXPANDED FIGURE 8

## E    EXPERIMENTAL DETAILS

### E.1    VIPAINT

**Choosing $(T_s, T_e)$ for VIPaint**    Extensive prior work Song et al. (2021b); Nichol & Dhariwal (2021); Dhariwal & Nichol (2021); Karras et al. (2022) explores different noise schedules for training diffusion models, and how it affects the generated image quality. Since we use these diffusion models incorporating different noise schedules, our latent hierarchical posterior needs to account for this shift, and we show that it is flexible to do so. To concentrate posterior inference on the noise levels which are most crucial to perceptual image quality, we define our posterior at intermediate time steps that induce a signal-to-noise-ratio $(\alpha_t^2/\sigma_t^2) \in [0.2, 0.5]$, Kingma et al. (2021b)) approximately across our experiments. This corresponds to choosing $(T_e = 5, T_s = 2)$] for the pixel-based EDM prior with a variance-exploding noise schedule and $(T_e = 550, T_s = 400)$ ((DDIM sampling coefficient, $\eta = 0.2$) ) for the LDM prior using the VP noise schedule for both LSUN and ImageNet256 datasets. VIPaint is not sensitive to any subset of $K$ timesteps in between this signal-to-noise range. For instance, for VIPaint-4, we take $[T_e = 5, 4, 3.5, 2.5, T_s = 2]$ for the EDM noise schedule and $[T_e = 550, 500, 450, T_s = 400]$ for the LDM prior.

**Choosing $K$**    We discussed in section **??** that for a $K$ step hierarchical posterior, the optimization run-time of VIPaint is O(K). From the experiments conducted, we see that $K$ can be easily selected to trade-off time and sample quality.

**Initialization**    We follow the forward and reverse diffusion process defined by each VE and VP noise schedules to initialize VIPaint's variational parameters. For LDM prior, we use the lower dimensional encoding of $y$. We provide a comprehensive summary in Table 2.

Table 2: Initialization of Variational Parameters for VE and VP Schedules

| VI Parameters | VP Schedule | VE Schedule |
|---|---|---|
| $\mu_{T_e} = \alpha_{T_e} y + a_1 \sigma_{T_e} \epsilon$ (Scale factor to retain information from $y$.) | $a_1 = 0.8$ | $a_1 = 0.01$ |
| $\mu_{s(i)} = \alpha_{s(i)} y + a_2 \sigma_{s(i)} \epsilon$ (Noise adding process is still quite high for VE schedules.) | $a_2 = 1$ | $a_2 = 0.01$ |
| $\tau_{T_e} = \sigma_{T_e}$ (From the forward diffusion process. ) | – | – |
| $\tau_{s(i)|s(i+1)}$ (From the reverse diffusion process.) | Eq. 26 with scaling factor $a_3/\eta$ $a_3 = 0.7$ | Eq. 23 |
| $\gamma_{s(i)} \forall i \in [1, K]$ (Weights samples from prior to construct plausible and close to real looking samples.) | 0.98 (ImageNet256), 0.88 (LSUN) | 0.5 |

**Optimization**    We fit three sets of variational parameters at every $i$-th critical time in our hierarchy: means, $\mu_{s(i)}$, variances $\tau_{s(i)}^2$ and weights $\gamma_{s(i)}$. Instead of optimizing $\tau$ and $\gamma$ directly, we optimize the real valued $\tilde{\tau} = \log \tau^2$, and $\tilde{\gamma} = \log(\frac{\gamma}{1-\gamma})$. We optimize this set of variational parameters $\lambda = \{\mu, \tilde{\gamma}, \tilde{\tau}\}$ using Adam with an initial learning rate of $\{0.1, 0.1, 0.01\}$ respectively and decreasing the learning rate by a factor of 0.99 every 10 iterations. We find this setting to be robust across all prior diffusion models and datasets in our work.

During pre-training, most diffusion models parameterize the mean prediction at every diffusion time step $t$ and fix variances, however some previous work Nichol & Dhariwal (2021); Dhariwal & Nichol (2021) has found that (with appropriate training "tricks") learning variances improves performance. Some previous works like ReSample tunes this as a hyperparameter. We instead learn this in our work, and we adjust learning rates to avoid local optima in this process. We optimize the parameters in VIPaint with $K = 2$ for 50 iterations; VIPaint with $K = 4$ is optimized for 100 steps in the case of LSUN Churches, 150 steps for the ImageNet64 dataset and 250 steps for the ImageNet256 dataset.

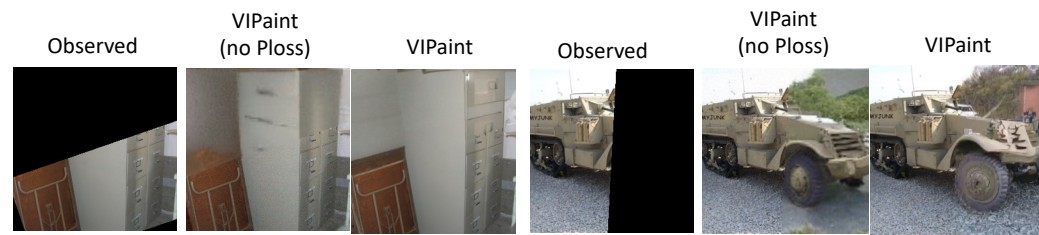

Figure 9: An ablation showing the effect of the addition the perceptual loss (PLoss) in the reconstruction term for the task of image inpainting using latent diffusion priors. We see that even though VIPaint can inpaint the image semantically without the Perceptual loss, this loss becomes important to produce sharper reconstructions.

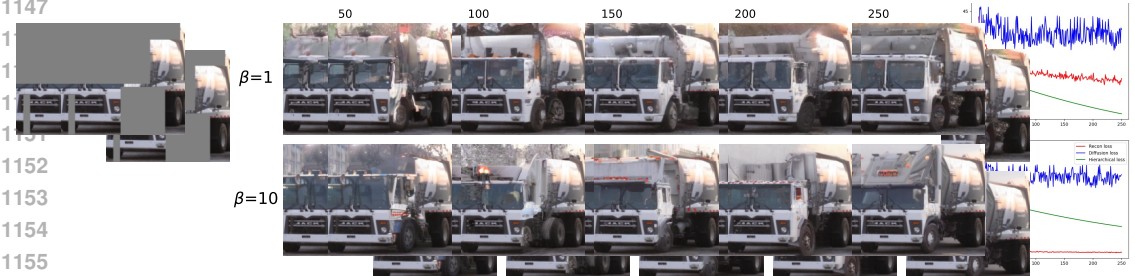

Figure 10: (**Left**) We show the effect of the hyperparameter $\beta$ with VIPaint with respect to optimization iterations. (**Right**) we show the respective loss curves. With $\beta = 10$, VIPaint captures more variations under the diffusion prior instead of "setting" to one kind of completion with $\beta = 1$.

**Sampling**  Post training, we take 400 iterative refinement steps from $T_s = 400$ in the LDM variance preserving schedule to sample inpaintings using a scale factor of 2, similar to the DPS algorithm using perceptual loss. On the other hand, for the EDM prior, we take 700 refinement steps to produce inpaintings after $T_s = 2$, with scale 5 (similar to DPS tuned for EDM in our work). This scale hyperparameter is tuned over the values [0.1, 0.5, 1, 2, 5, 10] on a validation set of 20 images. During the sampling phase, we use the classifier-free guidance rule with scale = 3 for the ImageNet256 latent diffusion prior.

**Reconstruction loss**  We assume $p(y|z_{T_s})$ as a Laplacian distribution, where the mean is given by $y$ and a scale parameter, which is computed over 100 images per dataset as a standard deviation over all pixel dimensions. For the 256 pixel datasets, this is 0.56, and for ImageNet64 it is 0.05. In addition to this, we add the perceptual loss for LDM priors, computing them via feeding the pre-trained Inception network with masked images and masked reconstructions. See Fig. 9 for the benefits of using the perceptual loss with VIPaint. Additionally, we use $\beta = 1$ for VIPaint with $K = 1$, which is optimized for 50 iterations for faster convergence. For VIPaint with $K = 4$, we use $\beta = 50$ for pixel-based EDM prior and $\beta = 10$ for LDM prior. We show the effect of the different $\beta$ values in Fig. 10. Generally speaking, higher values of $\beta$ explores the diffusion latent space more and lower values weighs the likelihood term relatively more and converges faster to a solution.

**Descretization of timesteps for prior diffusion loss** $L_{(T_e, T)}(z_{T_e})$  Lastly, we directly adapt the descretization technique from EDM Karras et al. (2022) to compute the diffusion loss. We use $\rho = 7$ across all models and datasets as used by Karras et al. (2022).

### E.2  BASELINE DETAILS

Across all the baselines applicable to the latent diffusion models for the ImageNet256 dataset, we use the classifier-free guidance with a scale 3 Rombach et al. (2022b).

**Blended**  We run blended for 1000 discretization steps using the EDM and LDM prior.

**RePaint**  RePaint uses a descritization of 256 steps along with the standard jump length = 10, and number of times to perform this jump operation also set to 10, following standard practice Lugmayr et al. (2022).

**DPS**  Similar to blended, we take 1000 denoising steps for DPS and set scale = 5 for the edm-based diffusion model, while take 500 steps and keep scale as 0.5 for the Latent Diffusion prior (similar to the original work in Chung et al. (2023)). When using the perceptual loss for the latent diffusion prior, we increase the scale to 2.

**PSLD**  This is an inference technique only for the Latent Diffusion prior. Similar to DPS, we take 500 steps and keep scaling hyperparameters set to 0.2 as opposed to choosing 0.1 in the original paper Rout et al. (2023). We observe artifacts in the innpainted image if we increase the scale further as also observed by Chung et al. (2024).

**CoPaint**  We directly adapt the author-provided implementation of CoPaint and CoPaint-TT Zhang et al. (2023) to use the EDM prior. Apart from the diffusion schedule and network architecture (taken from EDM) all other hyperparameters are preserved from the base CoPaint implementation.

**RED-Diff**  As with CoPaint, We directly adapt the author-provided implementation of RED-Diff Mardani et al. (2024) and Red-Diff (Var) to use the EDM prior. In this case we increased the prior regularization weight from 0.25 to 50, which we found gave improved performance and more closely matches our VIPaint settings.

**ReSample**  As with other baselines, we directly adapt the author-provided implementation of Re-Sample Song et al. (2024) for the LDM prior. Because the original code takes larger optimization steps, resulting in high sampling time, we decrease the number of optimization steps to 50, such that the wall-clock run-time of this method matches the other baselines.

## F  INFERENCE TIME.

We report the time taken for each inference method to produce 10 inpaintings for 1 test image. VIPaint with $K = 2$ is comparable to the baseline methods in terms of wall clock time and the number of functional evaluations (#NFEs) of the denoising network. Red-Diff, Blended and RePaint baseline methods do not take gradient of the noise prediction network, whereas all other methods require gradients. In terms of time and NFEs, we can see that VIPaint (fast) takes comparable time and NFEs as other baselines, but performs far better (Table **??**).

Overall, gradient based methods like DPS take longer with an LDM prior because of the use of a decoder per gradient step. PSLD additionally utilizes the encoder and hence, takes longer than DPS.

Table 3: Time (in mins) & #NFEs of denoising network per Inference method using EDM Prior (top) and LDM Prior (bottom)

| Red-Diff | Blended | DPS | RePaint | CoPaint | CoPaint-TT | VIPaint-2 | VIPaint |
|---|---|---|---|---|---|---|---|
| (1.13, 1000) | (1.13, 1000) | (2.55, 1000) | (2.8, 4700) | (2.6, 500) | (5.4, 1000) | 3.3 = (1.5, 150) (optimization) (1.8, 700) (sampling) | 11.8 = (10, 900) (optimization) (1.8, 700) (sampling) |

| Dataset | Blended | DPS | PSLD | VIPaint-2 | VIPaint |
|---|---|---|---|---|---|
| ImageNet256 | (4, 1000) | (10, 500) | (12.4, 500) | 10 = (2, 150) (optimization) (8, 400) (sampling) | 18 = (10, 1250) (optimization) (8, 400) (sampling) |
| LSUN | (1.3, 1000) | (5.1, 500) | (7.0, 500) | 6.4 = (2.1, 150) (optimization) (4.3, 400) (sampling) | 10 = (5.53, 750) (optimization) (4.3, 400) (sampling) |

## G  COMPUTATIONAL RESOURCES

All experiments were conducted on a system with 4 Nvidia A6000 GPUs.

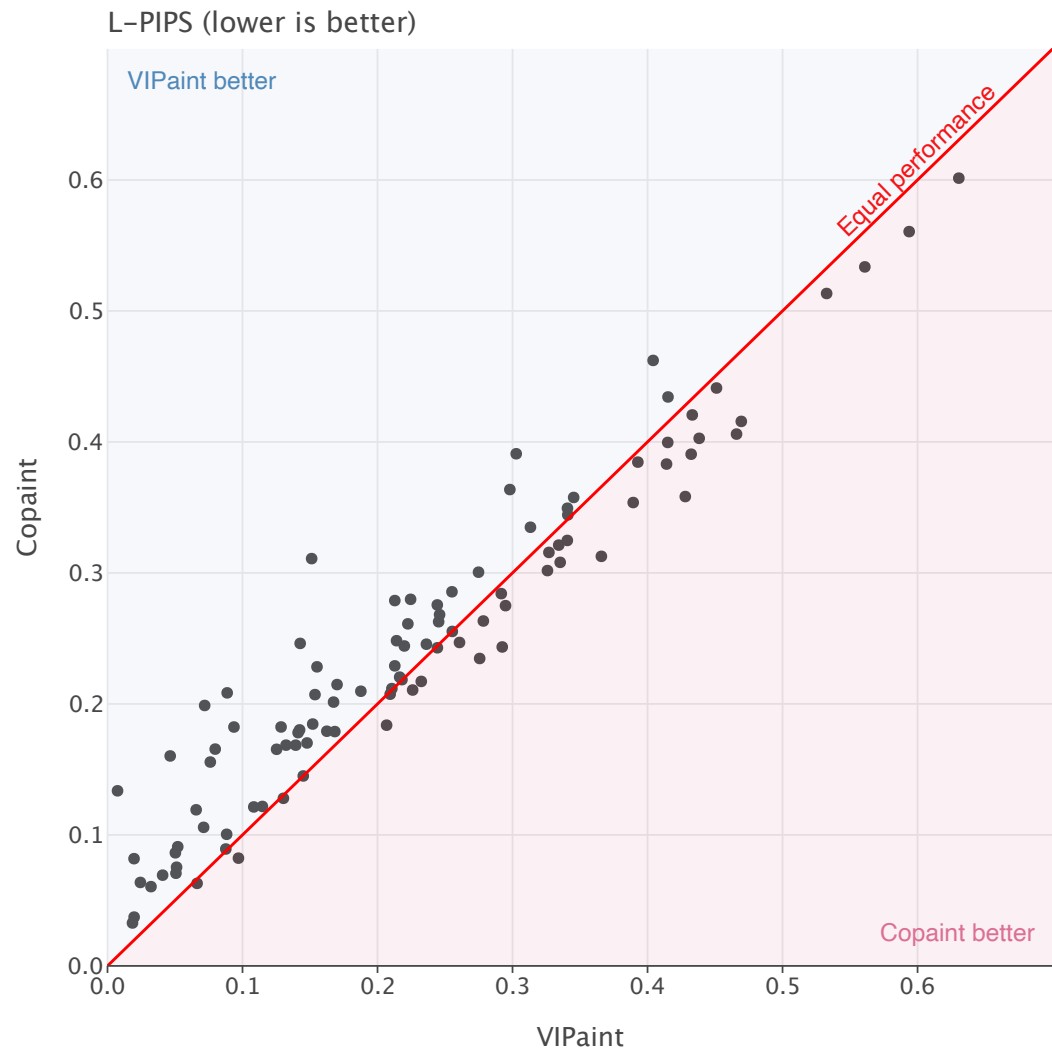

Figure 11: Paired comparison of LPIPS scores for VIPaint-2 and CoPaint with time-travel (CoPaint-TT) on the Imagenet64 "Random Mask" inpainting task (expanding on the experiment shown in table 1. Each point shows the mean LPIPS score across 10 sampled completions of the masked image, with the x and y coordinates showing the VIPaint and CoPaint-TT scores respectively. Additionally, we validated that VIPaint improves on CoPaint-TT using a one-sided paired t-test on the mean LPIPS scores of each method. We found that the improvement was statistically significant with a p-value of **4.133e-05**. As the normality assumption of the t-test may not hold, we also verfied the results using a nonparametric Wilcoxon signed ranked test, which indicated a statistically significant improvement with a p-value of **0.000114**

## H ADDITIONAL EXPERIMENTS

### H.1 ANALYSIS OF IMAGENET RESULTS

Fig. 11 shows details of the comparison between VIPaint and CoPaint with time-travel over 100 randomly selected images from the Imagenet-64 task.

### H.2 LINEAR INVERSE PROBLEMS

For linear inverse problems other than inpainting, we consider the following tasks: (1) Gaussian deblurring and (2) super resolution. For Gaussian deblurring, we use a kernel with size $61 \times 61$ with standard deviation 3.0. For super resolution, we use bicubic downsampling, similar setup as

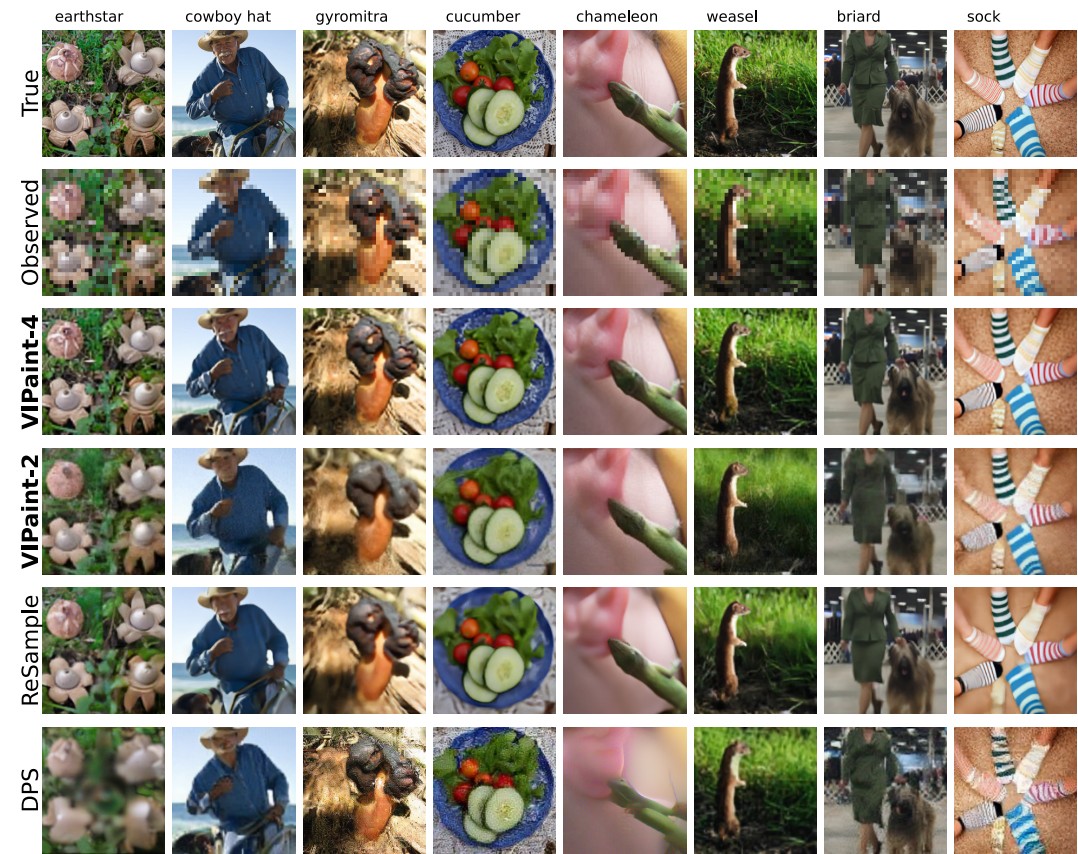

Figure 12: Qualitative results on Imagenet256 for Super Resolution. We see that DPS produces completely blurry images. We see improvements with ReSample. In contrast, VIPaint-4 leads to samples closer to the true image and produces *very* realistic images.

| Task | ImageNet256 | | | | ImageNet64 | |
| | Super-resolution 4x | | Gaussian Deblur | | Gaussian Deblur | |
| Metric | LPIPS ↓ | PSNR ↑ | LPIPS ↓ | PSNR ↑ | LPIPS ↓ | PSNR ↑ |
| --- | --- | --- | --- | --- | --- | --- |
| VIPaint-4 | **0.33** | **19.31** | 0.44 | 17.90 | **0.306** | 13.47 |
| VIPaint-2 | 0.46 | 16.36 | 0.48 | 16.35 | **0.305** | 13.60 |
| ReSample | 0.395 | 18.410 | **0.435** | **18.03** | – | – |
| PSLD | 0.67 | 7.77 | 0.583 | 0.022 | – | – |
| DPS | 0.579 | 12.99 | 0.595 | 12.608 | 0.319 | **13.43** |

Table 4: Quantitative results (LPIPS, PSNR) for solving linear inverse problems on ImageNet256 using LDM priors and ImageNet64 using EDM priors. Best results are in bold and second best results are underlined. For nonlinear deblurring.

Chung et al. (2023). Even though the focus of VIPaint is to remedy inconsistencies in image completion tasks, it can also be extended to linear inverse problems like Super Resolution and Gaussian Deblurring.

We compare the performance of VIPaint with ReSample, PSLD & DPS for ImageNet256 dataset using the LDM prior and for the pixel-based model, we include results for Gaussian Deblurring-compring VIPaint with DPS. Since the Peak-Signal-to-Noise-Ratio (PSNR) is well defined for such tasks, we report it along with LPIPS in Table 4. Some qualitative plots are in Fig. 12 and 13. A quantitative analysis is reported in Table **??** and qualitative results in Fig. 14 and **??**. We see that VIPaint shows strong advantages over ReSample, DPS and Red-Diff for complex image datasets like ImageNet.

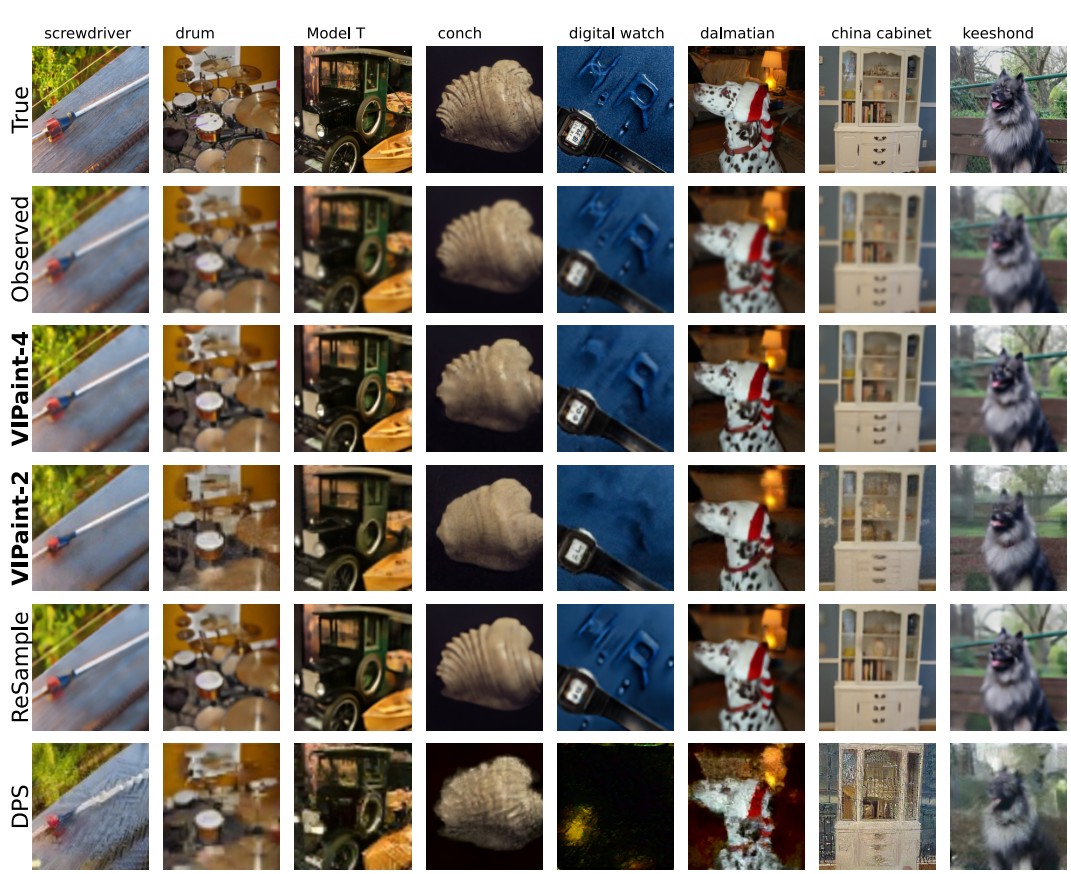

Figure 13: Qualitative results on Imagenet256 Gaussian DeBlurring using LDM prior.

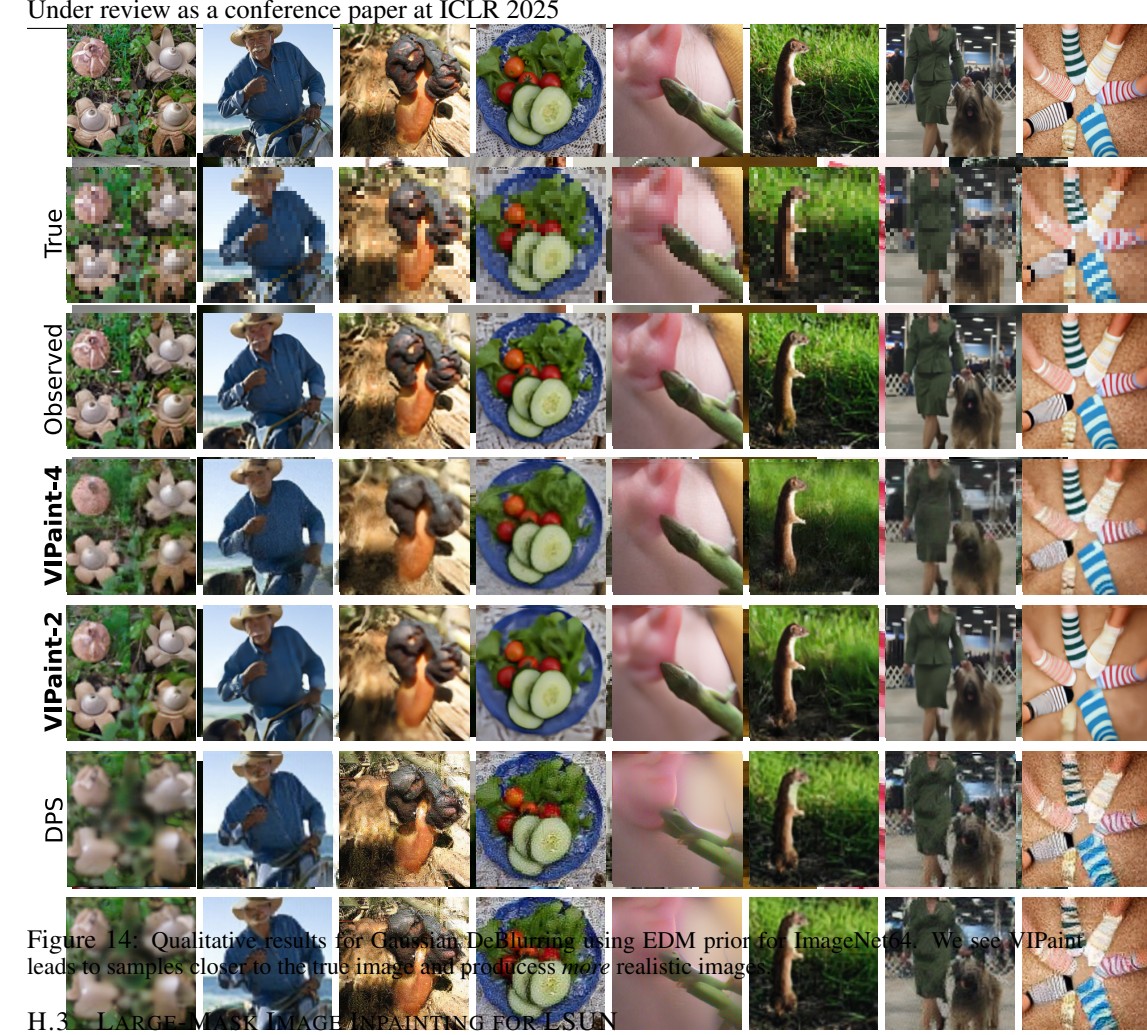

Figure 14: Qualitative results for Gaussian DeBlurring using EDM prior for ImageNet64. We see VIPaint leads to samples closer to the true image and produces *more* realistic images.

### H.3    LARGE-MASK IMAGE INPAINTING FOR LSUN

We show some qualitative figures for large masking ratios in Fig. 15 for LSUN-Church.

### H.4    SMALL-MASK IMAGE INPAINTING FOR LSUN, IMAGENET256

We show some qualitative figures for small masking ratios (upto 20% of the image is corrupted) in Fig. 16 and 17 for ImageNet-256 and LSUN-Church datasets respectively.

### H.5    VIPAINT CAPTURES MULTI-MODAL POSTERIOR

In addition to producing valid inpaintings, we show multiple samples per test image for all datasets we consider in Fig. 18-24.

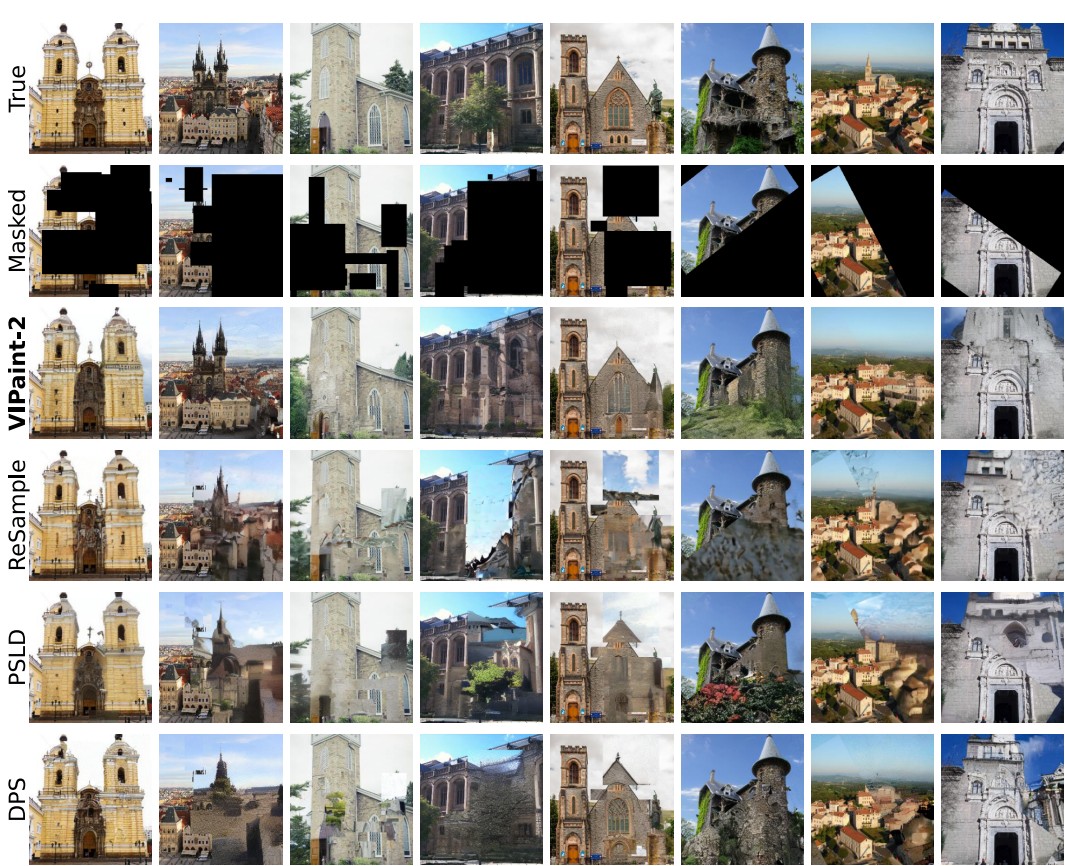

Figure 15: Qualitative results for LSUN-church dataset using LDM prior for the tasks of image inpainting with large masks. We see that VIPaint-2 can inpaint the images consistently and without any artifacts at the mask borders.

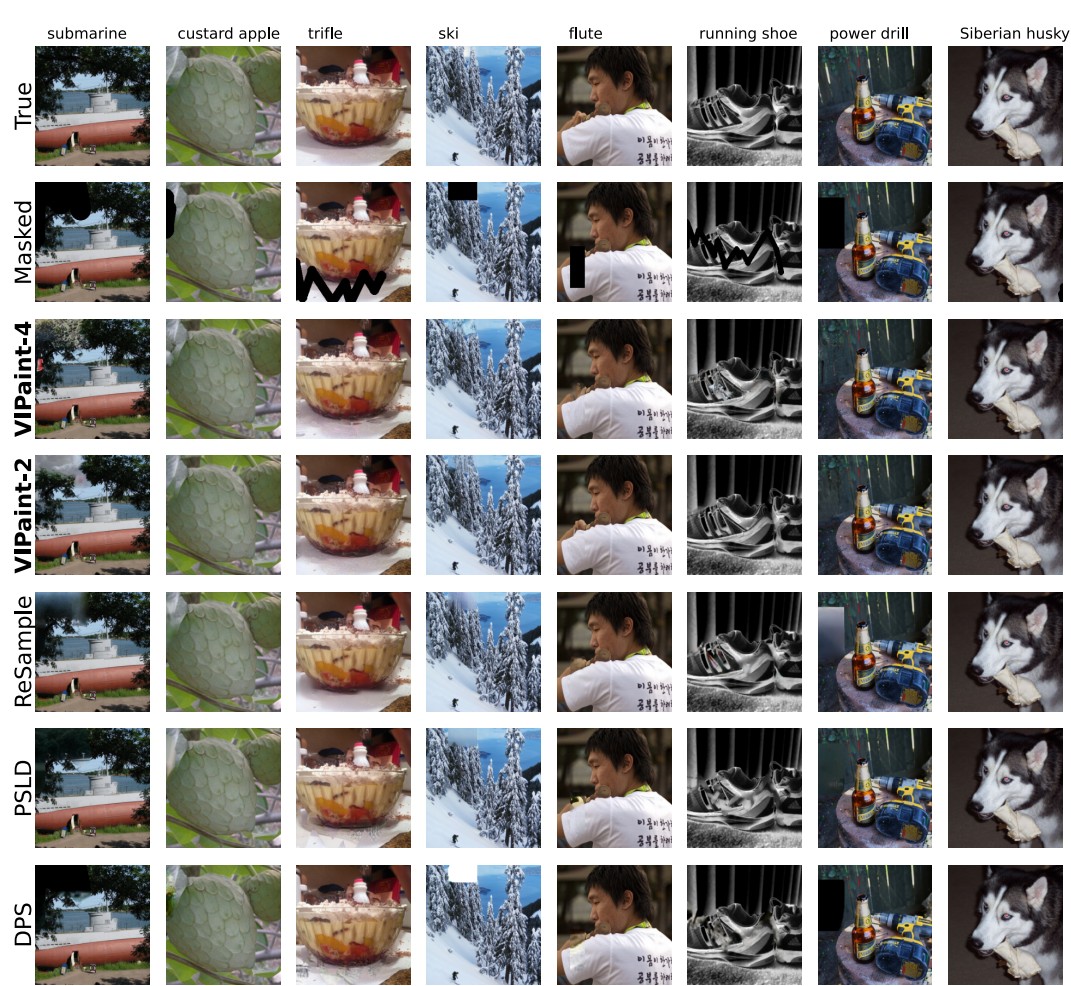

Figure 16: Qualitative results on the performance across methods for small masking ratios for ImageNet256 dataset using LDM prior. All methods seem to perform reasonably well in this regime.

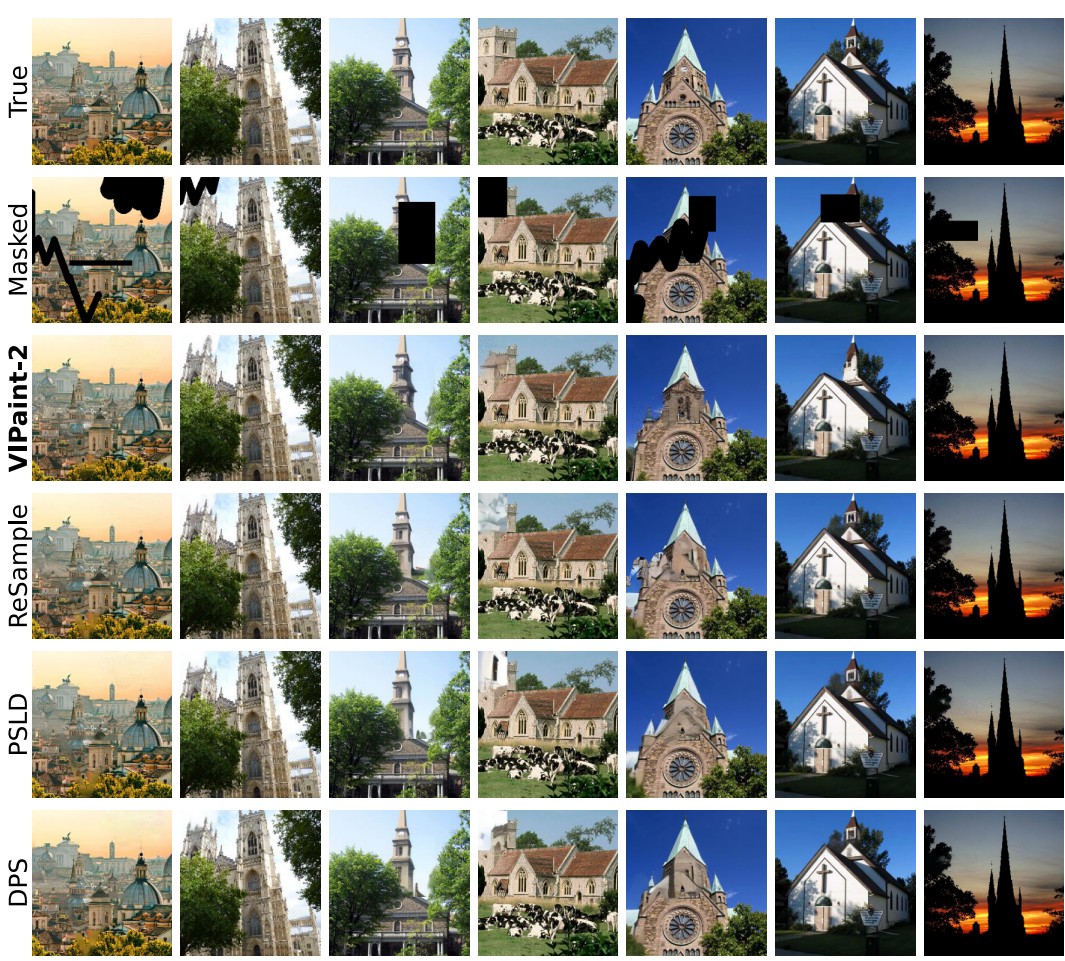

Figure 17: Qualitative results on the performance across methods for small masking ratios for LSUN-Church dataset using LDM prior. All methods seem to perform reasonably well in this regime. However, some minor artifacts start to show up for contiguous masks.

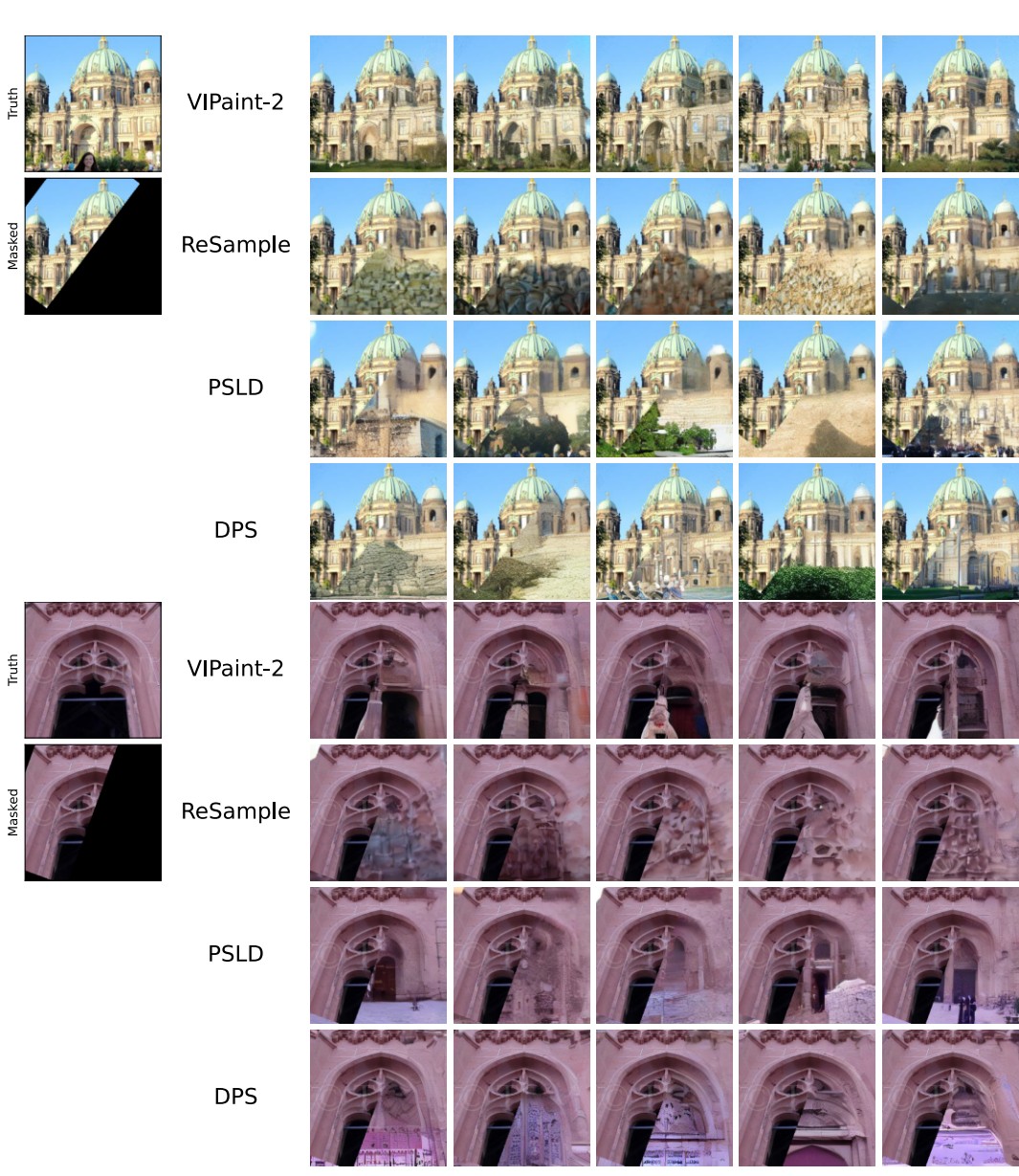

Figure 18: LSUN diversity results. Examples of diverse generation using VIPaint and baseline methods on LSUN using the same input and different initial noise.

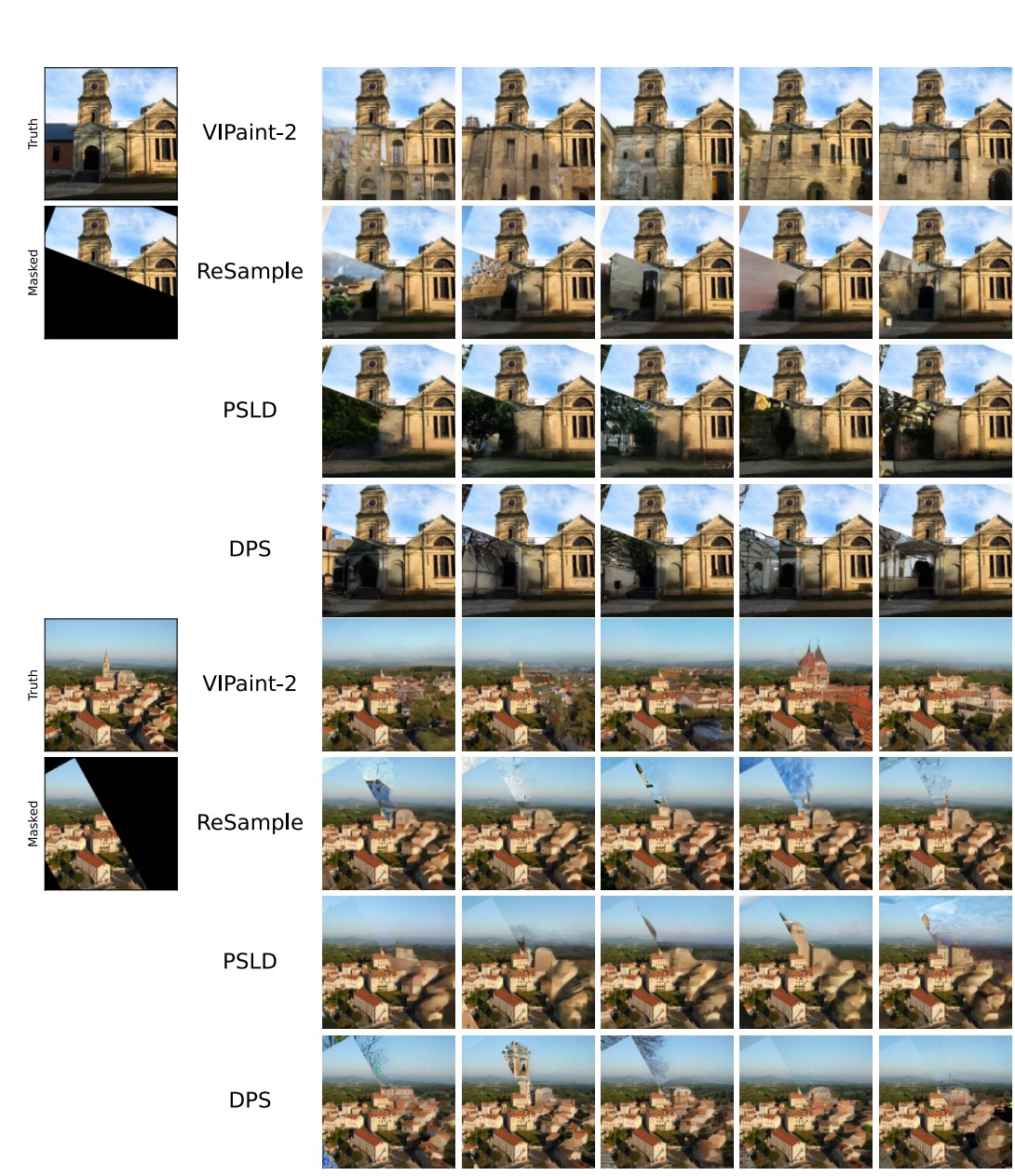

Figure 19: LSUN diversity results.

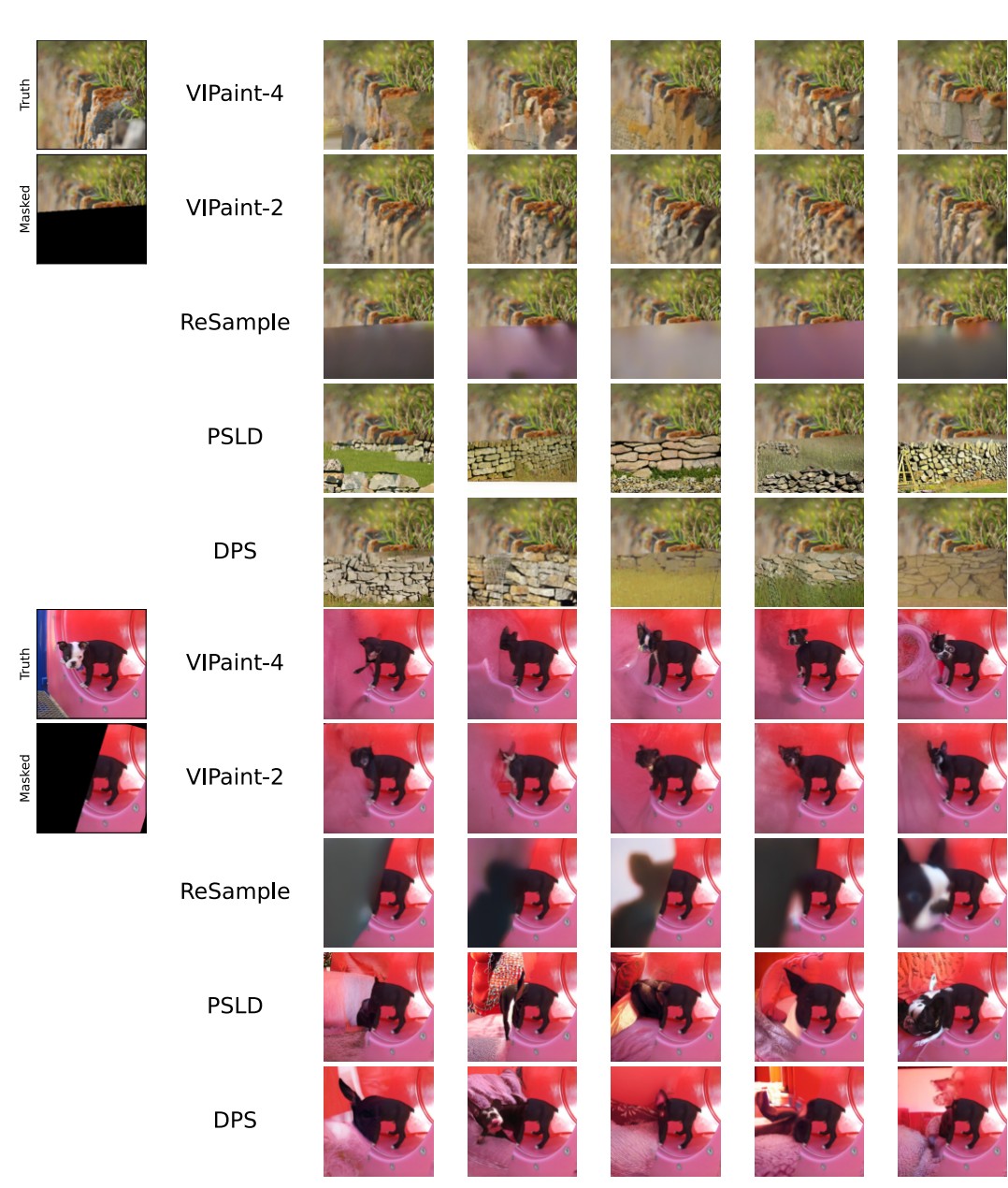

Figure 20: ImageNet-256 diversity results. Some examples of diverse generation using VIPaint and baseline methods on ImageNet using the same condition and masked input but with different initial noise.

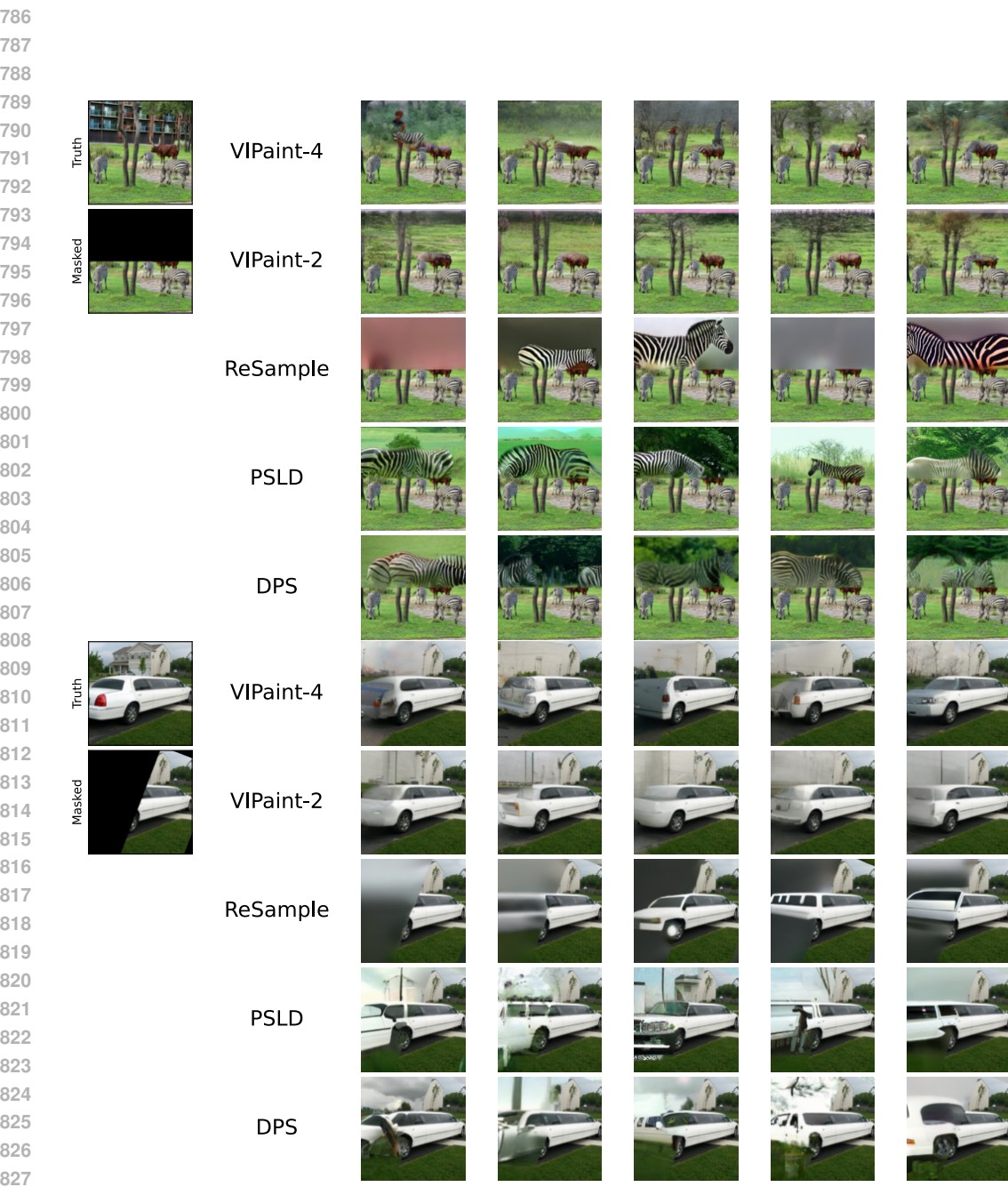

Figure 21: ImageNet diversity results.

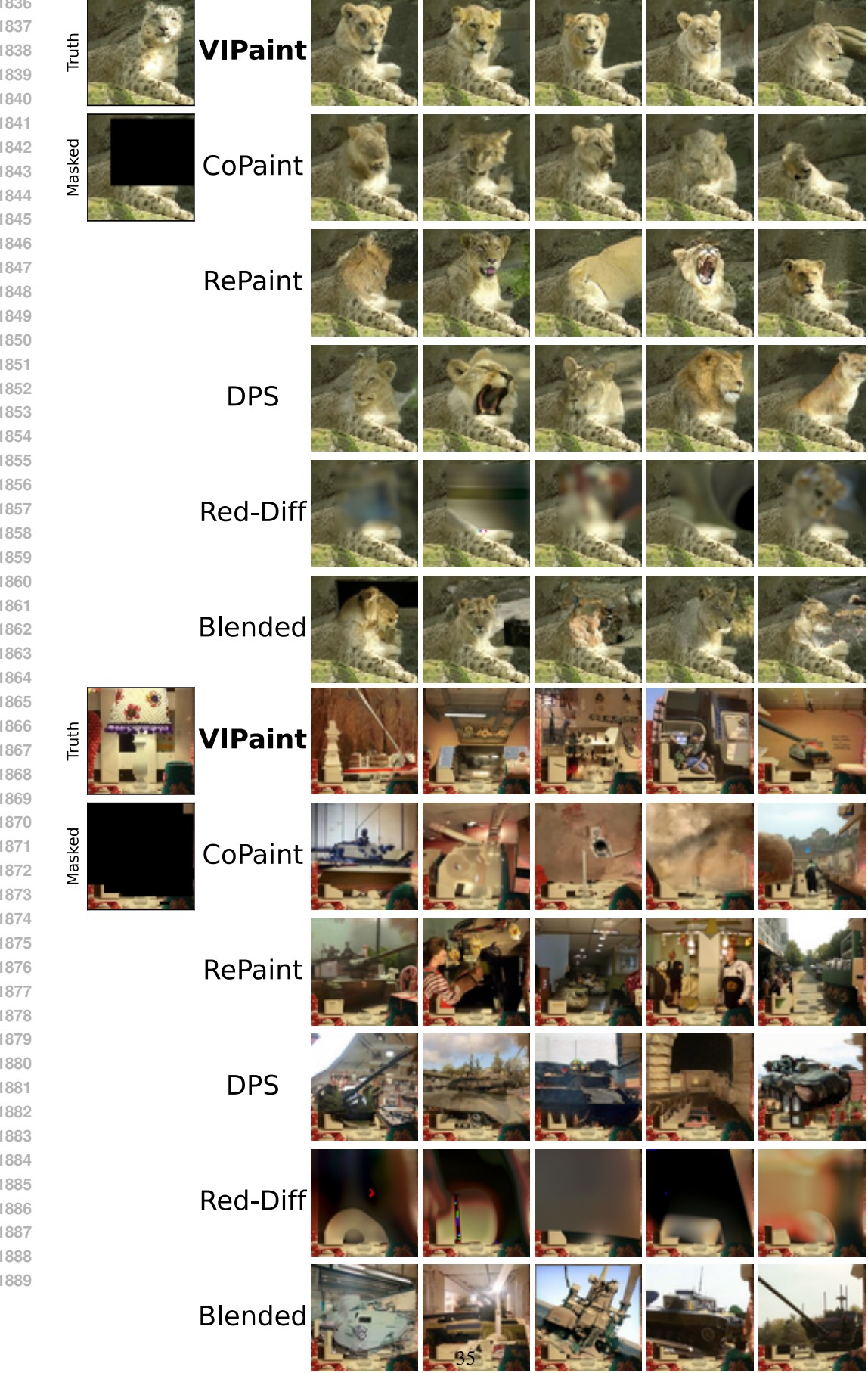

Figure 22: ImageNet64 diversity results with the same class condition but different initial noise.

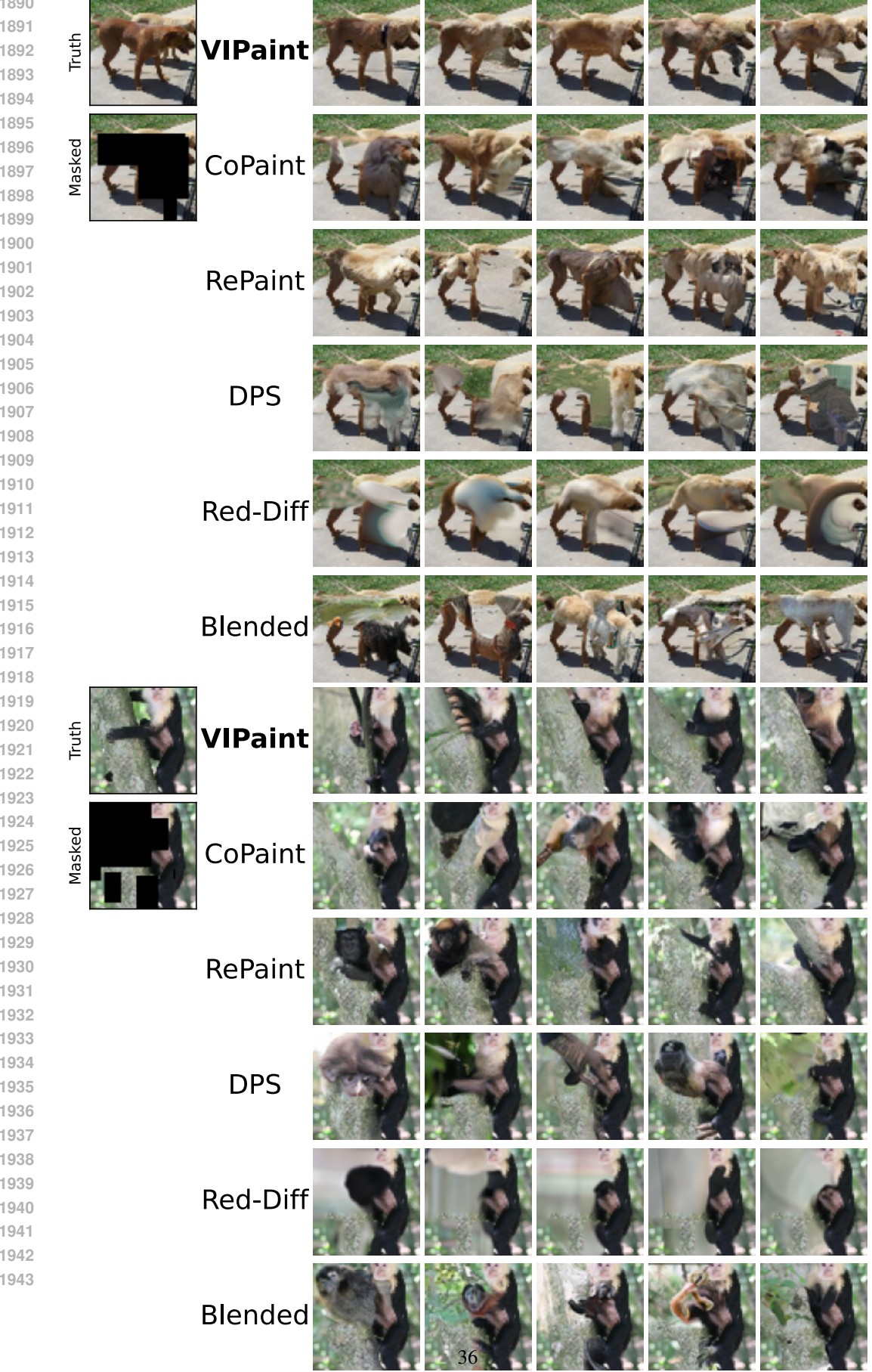

Figure 23: ImageNet64 diversity results with the same class condition but different initial noise.

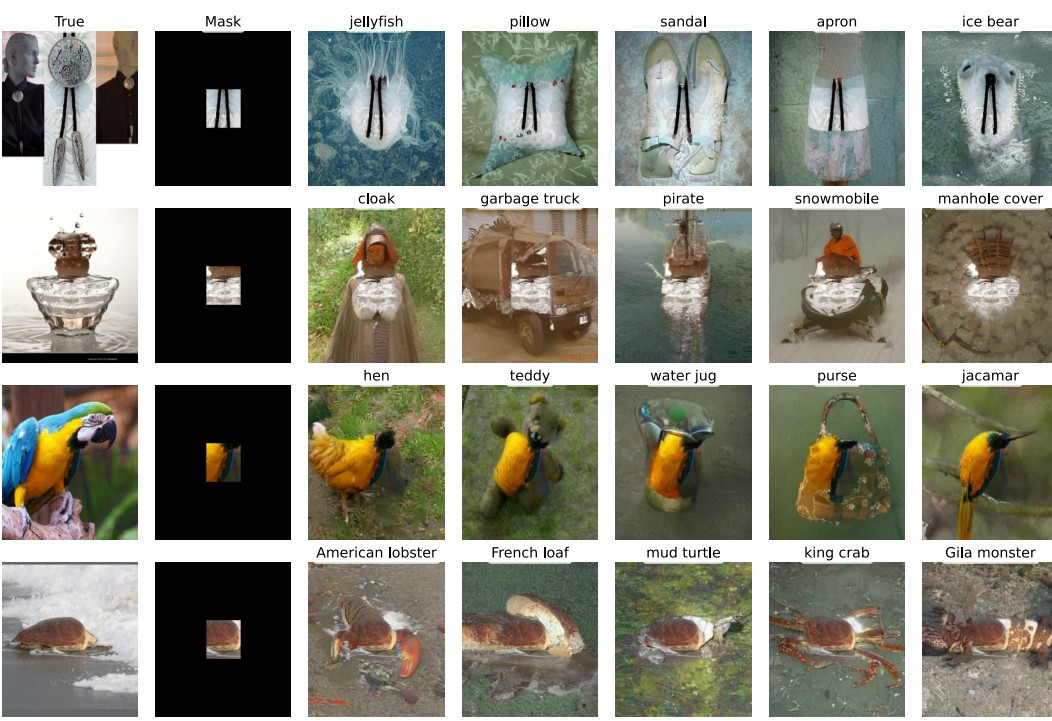

Figure 24: Qualitative results for VIPaint diversity for ImageNet256 with LDM prior using different class conditioning. We see that VIPaint follows the input label and ensures consistency with the observed set of pixels.