# OpenReview forum: "VIPaint: Image Inpainting with Pre-Trained Diffusion Models via Variational Inference"
_ICLR.cc/2025/Conference — ICLR 2025 Conference Withdrawn Submission_

### Official Review · Reviewer_iCq3 · 2024-10-28

**Soundness:** 1
**Presentation:** 1
**Contribution:** 1
**Rating:** 3
**Confidence:** 4

**Summary:**

Turning a general diffusion model into a posterior sampler for handling inverse problems requires guidance terms that introduce the measurement y into the synthesis process. This is a challenging task due to the annealing noise, forcing the proposed algorithms to adopt various approximation strategies. Doing so in the latent domain (e.g. for Stable-Diffusion) is even more challenging, due to the gap between the pixel and the latent domains.

This work belongs to this breed of techniques, focusing on the inpainting problem (although touching on super-res and deblurring as well), and operating in both the pixel and the latent domains. To quote the authors, they propose a "hierarchical variational inference algorithm that analytically marginalizes missing features, and uses a rigorous variational bound to optimize a non-Gaussian Markov approximation of the true diffusion posterior". What does that mean? In my opinion, NOTHING !!

**Strengths:**

1. The inpainting results are very good.
2. Sections 1-3 are well written and pleasant to read.

**Weaknesses:**

1. Section 4 in this paper, which introduces the proposed algorithm, is simply impossible to follow. You might think that reading the appendices will help, but it does not.
2. How this algorithm relates to RedDiff? DPS? PiGDM? One cannot tell from the (very poor) description of the algorithm.
3. How complex is the proposed algorithm? One cannot know because it is not clear what it does!! A pseudo-code could have been helpfull here.
4. What makes this method specifically tailored to inpainting? This is unclear at all.
5. In the experimental part, assessing inpainting results by LPIPS alone is simply insufficient. When missing parts are big, completing the image along the lines of the original one is simply not the proper approach. FID for evaluating the perceptual quality of the results and CRMSE to check the faithfullness of the given parts to the recovered ones would have been far better measures. In addition, showing the variance between various samples to show the spread could have been informative as well. Therefore, Table 1 is far from convincing. The same applies to Table 4 for the other inverse problems - FID (or precision/recall) is very much missing.

**Questions:**

None.

**Details Of Ethics Concerns:**

None.

---

### Official Review · Reviewer_8zhV · 2024-10-29

**Soundness:** 2
**Presentation:** 1
**Contribution:** 2
**Rating:** 3
**Confidence:** 4

**Summary:**

Solving inverse problems using pre-trained diffusion models is challenging because the noise conditional measurements are unknown. Existing methods aim to approximate this unknown quantity using first- or second-order Tweedie’s formula. In this paper, the authors propose a hierarchical optimization procedure to get around this issue. Instead of explicitly approximating this quantity, the authors construct a non-Gaussian Markov approximation of the true diffusion posterior. The main idea is an extension of Red-Diff to a noisy posterior. Extensive experiments on pixel and latent diffusion models show that the proposed method is competitive to prior works in random inpainting, box inpainting, super-resolution, and Gaussian deblur tasks.

**Strengths:**

1. Extending Red-Diff to a noisy posterior is a valuable idea and provides practical benefits, particularly when critical timesteps are *carefully* chosen in the reverse process.

2. Experimental results demonstrate improvements over prior works in tasks such as random inpainting, box inpainting, super-resolution, and Gaussian deblurring.

**Weaknesses:**

1. How do you identify which timesteps in the reverse process contain the critical relevant information? Can the authors provide more details on how they select the critical timesteps, or if there is any analysis showing the impact of different timestep choices?

2. Line 368-372: The main contribution of the paper is a straightforward extension of Red-Diff, except that in this paper the posterior is assumed to be well-defined for noisy latents (which is already a very strong assumption). Given access to such noisy posterior, even the existing inverse problem solvers like DPS, PSLD or Red-Diff will perform relatively better. The main issue is that the posterior is not well-defined for noisy latents. It is defined only for zeroth noise level, which justifies these prior works (e.g. Red-Diff). So, the main contribution of this paper needs to be clearly articulated in the context of these existing methods. The authors are encouraged to provide a more detailed comparison or analysis showing how their approach differs from and improves upon Red-Diff and other existing methods, particularly under the stronger assumption of noisy posterior.

3. The paper is poorly structured, spending excessive space on background information. The main contribution only appears in Section 4.2 on page 7, leaving insufficient room for experiments. As a result, experimental results are presented without adequate explanation, appearing as statements for completion rather than detailed analyses. The authors are encouraged to move parts of the background and related works from Section 2 and 3 to appendix to make room for more detailed explanation of their contributions and experimental results.

4. Typo in Line 1100, 1223, 1348: Missing reference labels make it difficult to follow which table or figure is referred to in this context.

5. The authors should clarify how their experimental settings differ from those in the original baselines [1,2,3]. It is essential to discuss the rationale for these changes and the potential impact on the results, especially because the baseline results here differ significantly from those reported in prior works [1,2,3].

6. What do the authors mean when they say in H.5 VIPaint captures multi-modal posterior? For a simple Gaussian mixture, the exchange of expectation in Tweedie’s formula has a bias that will prevent VIPaint from sampling the right posterior. The authors are requested to provide a more rigorous explanation or proof of how VIPaint captures multi-modal posteriors, particularly in light of the bias that comes with Tweedie's (first-order) formula.

7. The results in Fig 18-24 either look very similar or distorted, which do not verify the claim that VIPaint samples multi-modal posterior. Could the authors provide specific metrics or analyses that could demonstrate the multi-modality of the sampled posteriors, rather than relying solely on visual inspection of the figures.


8. VIPaint uses a significantly higher computational budget than the compared baselines, such as DPS or PSLD. For a fair evaluation, the proposed method should be compared with state-of-the-art LDM solvers [2,3] that utilize (nearly) comparable computational resources.


9. The theoretical discussion in Appendix B covers well-known material and lacks new insights. The main contributions of this paper are introduced only toward the end. The authors are encouraged to present their main contributions earlier in the paper to clearly distinguish their work from established literature.


### Reference:


[1] Nazemi, A., Sepanj, M. H., Pellegrino, N., Czarnecki, C., & Fieguth, P. (2024). Particle-Filtering-based Latent Diffusion for Inverse Problems. arXiv preprint arXiv:2408.13868.

[2] Chung, H., Ye, J. C., Milanfar, P., & Delbracio, M. Prompt-tuning Latent Diffusion Models for Inverse Problems. In Forty-first International Conference on Machine Learning.

[3] Rout, L., Chen, Y., Kumar, A., Caramanis, C., Shakkottai, S., & Chu, W. S. (2024). Beyond first-order tweedie: Solving inverse problems using latent diffusion. In Proceedings of the IEEE/CVF Conference on Computer Vision and Pattern Recognition (pp. 9472-9481).

**Questions:**

Please see the weaknesses above.

---

### Official Review · Reviewer_Ynoq · 2024-11-03

**Soundness:** 3
**Presentation:** 3
**Contribution:** 2
**Rating:** 3
**Confidence:** 5

**Summary:**

In the paper 'VIPaint: Image Inpainting with Pre-Trained Diffusion Models via Variational Inference', variational inference is used to rigorously estimate the posterior distribution in masked regions of images by employing previously trained pixel- or latent diffusion models. It is benchmarked against 5 recent inference-based inpainting methods on 100 test images drawn from 2 datasets (ImageNet64 and LSUN-Churches256) and shows superior results (LPIPS & KID). The authors perform 10 reconstructions per each test image resulting in a total of 1000 reconstructions

**Strengths:**

The work propose a variational bound to get a better estimate of the posterior and in this way provide a fundamental approach for inpainting instead of just using a heuristic.
The experiments show some advantage over some other methods selected by the authors.
The method applies to both diffusion models that work on image space and diffusion models that work in the latent space (LDM)
The steps added in the proposed method is in the same order like other approaches for inpainting

**Weaknesses:**

While the authors propose an interesting approach its validation is very limited due to the following reasons:
1. Weak experimental setup: The proposed method has been tested on only 100 test images. This makes the reported quantitative numbers not very reliable. Ideally, the quantitative metrics should be reported on much larger test dataset. Given that VIPaint takes around 2 mins to inpaint, it should be easy to test this method on larger and more diverse datasets. Providing additional experimental results will strengthen this paper a lot.
2. Comparison to recent methods: There is no comparison to recent methods such as https://openaccess.thecvf.com/content/CVPR2024/papers/Liu_Structure_Matters_Tackling_the_Semantic_Discrepancy_in_Diffusion_Models_for_CVPR_2024_paper.pdf https://openaccess.thecvf.com/content/CVPR2024/papers/Chen_Dont_Look_into_the_Dark_Latent_Codes_for_Pluralistic_Image_CVPR_2024_paper.pdf
3. Insufficient experiments: When looking at these works, it is clear that they compare many more datasets than what is being evaluated in the current work. Thus, it is very clear that the comparisons being made in this work are quite limited (see point 1).
4. Lack of ablation studies: VIPaint introduces many hyperparameters in its variational loss objective. I looked at appendix E.1 and it only explains what was selected. The choices of hyperparameters seem very arbitrary and it is unclear how to select optimal values of these hyperparameters, and how the metrics vary with these hyperparameters. How can I know that the parameters selected translate beyond what is being shown in the paper.
5. Diffusion models used: If the model applies to LDM why not show stable diffusion? I think this is very much to be expected given the claim of the authors.

**Questions:**

1. Why you don't demonstrate your method with stable diffusion?
2. This work introduces many hyperparameters, each of which is optimized with different learning rate. It is unclear if the method is robust to the choices of hyperparameters. Overall, it might be tricky to adapt this method to new datasets.
3. Why you don't compare to more recent works and larger datasets?
4. Why you don't perform ablation studies justifying the selected parameters and demonstrating the robustness of the method to the parameters (is it really robust to them? )

---

### Note · Authors · 2024-12-09

I have read and agree with the venue's withdrawal policy on behalf of myself and my co-authors.